# On Noise Abduction for Answering Counterfactual Queries: A Practical Outlook

**Saptarshi Saha**[*]                                                  *saptarshi.saha_r@isical.ac.in*
*Computer Vision and Pattern Recognition Unit*
*Indian Statistical Institute, Kolkata*

**Utpal Garain**                                                  *utpal@isical.ac.in*
*Computer Vision and Pattern Recognition Unit*
*Indian Statistical Institute, Kolkata*

**Reviewed on OpenReview:** *https://openreview.net/forum?id=4FU8Jz1Oyj&referrer=%5BTMLR%5D*

## Abstract

A crucial step in counterfactual inference is abduction - inference of the exogenous noise variables. Deep Learning approaches model an exogenous noise variable as a latent variable. Our ability to infer a latent variable comes at a computational cost as well as a statistical cost. In this paper, we show that it may not be necessary to abduct all the noise variables in a structural causal model (SCM) to answer a counterfactual query. In a fully specified causal model with no unobserved confounding, we also identify exogenous noises that must be abducted for a counterfactual query. We introduce a graphical condition for noise identification from an action consisting of an arbitrary combination of hard and soft interventions. We report experimental results on both synthetic and real-world German Credit Dataset, showcasing the promise and usefulness of the proposed exogenous noise identification.

## 1 Introduction

"What if?" questions are frequent in the decision-making system in almost all realms of knowledge. These questions evoke hypothetical conditions, usually contradicting factual evidence. For example, when a patient dies in the hospital, a natural question is: What would have happened if the clinicians acted differently? Another example is that had the candidate been male instead of female, would the decision from the admissions committee be more favorable? By and large, counterfactuals are key ingredients that go into explaining why things happened as they did. It is not possible to answer those questions using statistical tools only, but the method of counterfactual inference of hypothetical scenarios can prove helpful in those cases (Pearl, 2016).

Counterfactual techniques have been proposed into deep learning only in recent times (Schölkopf, 2019). For instance, there are inquisition in fairness (Kusner et al., 2017), recourse (Karimi et al., 2021), harm (Richens et al., 2022), mitigating bias in image classifiers (Dash et al., 2022), mitigating language bias in VQA (Niu et al., 2021), Zero-Shot Learning and Open-Set Recognition (Yue et al., 2021), mental health care (Marchezini et al., 2022).

The structural causal model (SCM) is the standard framework for computing the answers to the counterfactual queries. An SCM takes two sets of variables - exogenous and endogenous, and a set of structural assignments into account that assigns each endogenous variable a value according to the values of some other variables in the model. The exogenous variables are external to the model. We chose not to elucidate how they are caused. Each endogenous variable is a descendant of an exogenous variable. One can use the structural assignments to accurately compute the value of endogenous variables from the values of the exogenous variables. The SCM paradigm provides a three-step procedure for answering counterfactual

---

[*]first author

questions: Abduction, Action, and Prediction. Abduction is the tractable inference of the exogenous noise variables. Action is to perform interventions. Prediction is to compute the quantities of interest. Deep Learning approaches founded on these three steps have been recently introduced for generating counterfactuals. For instance, Pawlowski et al. (2020) employ normalizing flows and variational inference for enabling tractable counterfactual inference, Sanchez & Tsaftaris (2022) use diffusion models for counterfactual estimation, Axel Sauer (2021) proposes counterfactual generative networks, Dash et al. (2022) incorporates a structural causal model (SCM) in a variant of Adversarially Learned Inference for generating counterfactual images. Normalizing flow-based methods for answering counterfactual queries has received a lot of attention in no time. For example, Pawlowski et al. (2020)'s work on healthy magnetic resonance images of the brain has been extended to account for the clinical and radiological phenotype of multiple sclerosis (MS) by Reinhold et al. (2021). Wang et al. (2021) perform counterfactual inference to achieve harmonization of brain imaging data with different protocols and from different sites in a clinical study.

From a deep learning perspective, an exogenous variable might be considered as an inferred latent variable. To infer the state of the latent noise attached to an endogenous variable, we typically model a normalizing flow, perform amortized variational inference (in the case of very high dimensional variables) (Pawlowski et al., 2020) or use deterministic forward diffusion(Sanchez & Tsaftaris, 2022). Our ability to infer a latent variable comes at a computational cost as well as a statistical cost. To illustrate, the framework for counterfactual estimation by inferring exogenous noises via normalising flows parameterizes each structural assignment of an SCM as an invertible mechanism. Each mechanism explicitly calculates its inverse to enable efficient abduction of exogenous noises. These invertible architectures are typically computationally heavy. For a description of normalizing flows, see Appendix A and Papamakarios et al. (2019).

However, given an SCM, in practice, we are interested in counterfactual queries involving a few variables (not all)! For example, Reinhold et al. (2021) studied what the brain image of the subject would look like if the subject did not have lesions, given the observation that they have a 60 mL lesion load. While the proposed SCM consists of age, lesion volume of the subject, duration of MS symptoms, slice number, brain volume, biological sex, image, ventricle volume, and the expanded disability severity score. Hence, it is quite natural to ask for noise variables that we can get rid of from abducting. While Pawlowski et al. (2020) have mentioned (on a footnote) in the case of brain imaging example that abduction of the noise attached to 'sex' is not necessary as 'sex' has no causal parents in the SCM[1] (Figure 5, Pawlowski et al. (2020)), we are unaware of any dedicated effort to identify the noises that must be abducted to answer a counterfactual query.

In this context, our work shows that it may not be necessary to infer all the noise variables in the SCM and identifies exogenous noise variables that we must infer to answer a counterfactual query in a fully specified causal model with no unobserved confounding. We also introduce a graphical condition for noise identification from an action consisting of an arbitrary combination of hard, soft, and semi-soft (semi-hard) interventions. We report experimental results on both synthetic and real-world German Credit Dataset, showcasing the promise and usefulness of the proposed exogenous noise identification. The code for reproducing the results is available at https://github.com/Saptarshi-Saha-1996/Noise-Abduction-for-Counterfactuals.

## 2 Preliminaries

### 2.1 Background on structural causal models

A structural causal model(SCM) is defined as a tuple $\mathfrak{C} := (\mathcal{S}, \mathbb{P}(\boldsymbol{\epsilon}))$, where $\mathcal{S} = (f_1, f_2, ..., f_p)$ is a collection of $p$ deterministic structural assignments,

$$X_j := f_j(\mathbf{Pa}_j, \boldsymbol{\epsilon}_j), \quad j = 1, 2, .., p, \tag{1}$$

where $\mathbf{Pa}_j \subseteq \{X_1, ..., X_p\} \setminus \{X_j\}$ is the set of parents of $X_j$ (its direct causes) and $\mathbb{P}(\boldsymbol{\epsilon}) = \prod_{i=1}^{p} \mathbb{P}(\boldsymbol{\epsilon}_i)$ is the joint distribution over mutually independent exogenous noise variables. The graph of a structural causal model $\mathfrak{G}$ is obtained simply by drawing directed edges pointing from causes to effects. As assignments

---

[1]This need not be the case always. For instance, example 1(d).

are assumed acyclic, the directed graph $\mathfrak{G}$ induced by the SCM $\mathfrak{C}$ is also acyclic. Every SCM $\mathfrak{C}$ entails a unique joint distribution $P_{\mathbf{X}}^{\mathfrak{C}}$ over the variables $\mathbf{X} = (X_1, ..., X_p)$. The graph structure, along with the joint independence of the exogenous noises factorizes the entailed distribution $P_{\mathbf{X}}^{\mathfrak{C}}$ canonically into causal conditionals,

$$P_{\mathbf{X}}^{\mathfrak{C}}(\mathbf{X} = \mathbf{x}) := \mathbb{P}_{\mathfrak{G}}(\mathbf{x}) = \prod_{j=1}^{p} \mathbb{P}(x_j | \mathbf{pa}_j^{\mathfrak{G}}). \tag{2}$$

It is referred as causal (or disentangled or Markov) factorization. This allows to use $\mathfrak{G}$ for predicting the effects of interventions, defined as substituting one or multiple of its structural assignments, written as '$do(\cdots)$'. An intervention on a set of variables $\{X_t : t \in I\}$ is defined as substituting the respective structural assignments by

$$X_t := \tilde{f}_t(\widetilde{\mathbf{Pa}}_t, \tilde{\epsilon}_t), \quad t \in I.$$

The entailed distribution in the new SCM $\tilde{\mathfrak{C}}$ is called as intervention distribution, denoted by $P_{\mathbf{X}}^{\tilde{\mathfrak{C}}}$. The set of exogenous variables $\{\epsilon_t : t \notin I\} \cup \{\tilde{\epsilon}_t : t \in I\}$ in $\tilde{\mathfrak{C}}$ are required to be mutually independent. An intervention, where the structural assignment for a variable is modified by changing the function or the noise term, resulting in a change in the conditional distribution given its parents, is called soft/imperfect intervention. It is written as $do(X_t := \tilde{f}_t(\widetilde{\mathbf{Pa}}_t, \tilde{\epsilon}_t))$ (Peters et al., 2017). As the new SCM $\tilde{\mathfrak{C}}$ should have an acyclic graph, the set of allowed interventions thus depends on the graph $\mathfrak{G}$, induced by $\mathfrak{C}$. In this paper, we mainly focus on interventions with $\widetilde{\mathbf{Pa}}_t$ equals $\mathbf{Pa}_t$ or empty (that will be clear from the context). We use $\widetilde{\mathbf{Pa}}_t$ for a different purpose described in section 3. Independent Causal Mechanisms (ICM) Principle (Peters et al. (2017)) says that performing an intervention upon one mechanism $\mathbb{P}(X_i | \mathbf{Pa}_i)$ does not change any of the other mechanisms $\mathbb{P}(X_j | \mathbf{Pa}_j)(i \neq j)$. As a consequence, we get

$$P_{\mathbf{X}}^{\tilde{\mathfrak{C}}}(\mathbf{X} = \mathbf{x}) := \mathbb{P}_{\tilde{\mathfrak{G}}}(\mathbf{x}) = \prod_{j \notin I} \mathbb{P}_{\mathfrak{G}}(x_j | \mathbf{pa}_j^{\mathfrak{G}}) \prod_{j \in I} \mathbb{P}_{\tilde{\mathfrak{G}}}(x_j | \widetilde{\mathbf{pa}}_j^{\tilde{\mathfrak{G}}}). \tag{3}$$

When $\tilde{f}(\mathbf{Pa}_t, \tilde{\epsilon}_t)$ puts a point mass on a real value $a$, i.e., $\mathbb{P}_{\tilde{\mathfrak{G}}}(x_t | \mathbf{pa}_t) = \mathbf{1}_{x_t = a}$, we simply written it as $do(X_t = a)$ and call this an atomic/hard/perfect intervention. In particular, such constant reassignment disconnects $X_t$ from all its parents and imparts a direct manipulation disregarding its natural causes.

## 2.2 Counterfactuals

Given an observed outcome, counterfactuals are hypothetical retrospective interventions (cf. potential outcome): 'Given that we observed $(X_i, X_j) = (x_i^{obs}, x_j^{obs})$, what would $X_i$ have been if $X_j$ were $x_j'$? By assumption, the state of any observable variable is fully determined by the exogenous noises and structural assignments/equations. The unit-level counterfactual is defined as the solution for $X_i$ for a given situation $\epsilon = \epsilon$, where the equation for $X_j$ is replaced with $X_j = x_j'$. We denote it by $X_{i_{X_j \leftarrow x_j'}}(\epsilon)$ (Read: "The value of $X_i$ in situation $\epsilon$, had $X_j$ been $x_j'$"). We might be able to answer unit-level (or individual-level) counterfactual queries if we know the specific functional form of these structural equations. Mathematically, counterfactual inference can be formulated as three-step algorithm (Pearl (2009)):

1. **Abduction:** Predict the exogenous noise $\epsilon$ from the observations $\mathbf{x}^{obs}$, i.e., infer $\mathbb{P}(\epsilon | \mathbf{X} = \mathbf{x}^{obs})$.

2. **Action:** Perform interventions (e.g. $do(X_j = x_j')$) corresponding to the desired manipulations, resulting in a modified SCM $\tilde{\mathfrak{C}} := \mathfrak{C}|_{\mathbf{X} = \mathbf{x}^{obs}; do(X_j = x_j')} = (\tilde{S}, \mathbb{P}(\epsilon | \mathbf{X} = \mathbf{x}^{obs}))$, where $\tilde{S}$ is the collection of structural assignments modified by interventions.

3. **Prediction:** Compute the quantities of interest (e.g. $X_{i_{X_j \leftarrow x_j'}}(\epsilon)$) based on the distribution entailed by the counterfactual SCM $\tilde{\mathfrak{C}}$, denoted by $P_{\mathbf{X}}^{\tilde{\mathfrak{C}}} = P_{\mathbf{X}}^{\mathfrak{C}|_{\mathbf{X} = \mathbf{x}^{obs}; do(X_j = x_j')}}$.

The updated noise distribution of exogenous variables $\mathbb{P}(\epsilon | \mathbf{X} = \mathbf{x}^{obs})$ need not be mutually independent anymore. It is not always possible to determine the counterfactuals with probability 1. When we can't solve

for $\epsilon_i$ (e.g. function $f_i$ that maps $\epsilon_i$ to $X_i$ for a fixed value of $\mathbf{x}$ isn't invertible in noise term? ), we assume some prior distribution for $\epsilon_i$ and update $\mathbb{P}(\epsilon_i)$ by observations $\mathbf{x}^{obs}$ to obtain $\mathbb{P}(\epsilon_i|\mathbf{x}^{obs})$ (**Abduction**). In general, using Bayes' theorem,

$$\mathbb{P}(\epsilon = \epsilon | \mathbf{X}(\epsilon) = \mathbf{x}^{obs}) = \frac{\mathbf{1}_{\mathbf{X}(\epsilon)=\mathbf{x}^{obs}}\mathbb{P}(\epsilon = \epsilon)}{\sum_{\{\epsilon'|\mathbf{X}(\epsilon')=\mathbf{x}^{obs}\}}\mathbb{P}(\epsilon = \epsilon')}. \tag{4}$$

$\mathbf{X}(\epsilon)$ emphasizes that every endogenous variable $X_i$ is a function of $\epsilon$. In the case of non-invertible structural assignments, we do not get all the probabilities concentrated on one particular value of counterfactual $X_{i_{X_j \leftarrow x'_j}}(\epsilon)$; instead, we get a distribution. Averaging over the space of $\epsilon$, a potential outcome $X_{i_{X_j \leftarrow x'_j}}(\epsilon)$ induces a random variable that is simply denoted as $X_{i_{X_j \leftarrow x'_j}}$. The counterfactual distribution $\mathbb{P}(X_{i_{X_j \leftarrow x'_j}} = X_{i_{X_j \leftarrow x'_j}}(\epsilon)|X_j = x_j^{obs}, X_i = x_i^{obs})$ denotes the probability that $X_{i_{X_j \leftarrow x'_j}}$ is equal to the value $X_{i_{X_j \leftarrow x'_j}}(\epsilon)$ if $X_j$ is changed to a different value $x'_j$, given a specific observation $X_i = x_i^{obs}$ and $X_j = x_j^{obs}$. Let $\epsilon = \epsilon$ be one of the situation that leads to the observation $\mathbf{X} = \mathbf{x}^{obs}$ (more specifically, $X_i = x_i^{obs}, X_j = x_j^{obs}$). Then, in particular,

$$\mathbb{P}(X_{i_{X_j \leftarrow x'_j}} = X_{i_{X_j \leftarrow x'_j}}(\epsilon)|X_j = x_j^{obs}, X_i = x_i^{obs}) = \mathbb{P}(\epsilon = \epsilon|\mathbf{X} = \mathbf{x}^{obs}).$$

It advances us from unit-level counterfactual to population-level counterfactual that is not specific to a particular situation $\epsilon$ (but all the situations are considered, i.e., population), e.g., $\mathbb{E}(X_{i_{X_j \leftarrow x'_j}}|\mathbf{X} = \mathbf{x}^{obs})$. Expectation is taken over the whole population. $\mathbb{P}(\epsilon)$ defines a probability distribution over endogenous variables $\mathbf{X}$,

$$\mathbb{P}(X_i = x_i) = \sum_{\{\epsilon|X_i(\epsilon)=x_i\}} \mathbb{P}(\epsilon = \epsilon).$$

The probability of counterfactual statements is defined in the same manner, e.g.,

$$\begin{aligned}
\mathbb{P}(X_{i_{X_j \leftarrow x'_j}} = x'_i|X_j = x_j^{obs}, X_i = x_i^{obs}) &= \sum_{\{\epsilon|X_{i_{X_j \leftarrow x'_j}}(\epsilon)=x'_i\}} \mathbb{P}(\epsilon = \epsilon|\mathbf{X} = \mathbf{x}^{obs}) \\
&= \sum_{\epsilon} \mathbb{P}(X_{i_{X_j \leftarrow x'_j}}(\epsilon) = x'_i)\mathbb{P}(\epsilon = \epsilon|\mathbf{X} = \mathbf{x}^{obs})
\end{aligned} \tag{5}$$

With the help of such formulation, we are allowed to compute joint probabilities of every combination of counterfactual and observable events. Natural direct and indirect effects in mediation analysis, probability of necessity, probability of sufficiency (Pearl, 2016), harm (Richens et al., 2022), etc. are a few examples of counterfactual quantities.

### 2.2.1 Identifiability of counterfactuals

One of the fundamental questions in the counterfactual analysis is the question of identification: Can the counterfactual quantities be estimated from either observational or experimental data or both observational and experimental data? In a fully specified causal model, i.e., if all parameters of the causal model are known (including $\mathbb{P}(\epsilon)$), every counterfactual is identifiable and can be computed using the three steps - abduction, action, and prediction. Counterfactual quantities may not be generally identifiable even if we have interventional and observational distributions (Peters et al., 2017). For computing unit-level counterfactuals, one needs parametric forms of structural assignments. Often, in reality, we do not know the structural assignments and distributions of exogenous noises. Flow-based SCMs use normalizing flows to parameterize each structural assignment of an SCM as an invertible mechanism[2] and also make assumptions on the distributions of noises. One may not require parametric forms of structural equations to answer population-level counterfactuals. See Malinsky et al. (2019) for general identification of counterfactual quantities.

---

[2]We assume invertibility in the noise argument.

### 2.2.2 Scope of interventions for counterfactual analysis

Standard tools of the SCM framework do not inherently restrict intervention. One could, at least in theory, intervene unconditionally on any subset of variables to perform counterfactual analysis. Thus the choices of form and feasibility in the scope of interventions are delegated to the individual and\or the institution and made based on a semantic understanding of the modelled variables. For example, $Z$ can not be intervened in causal graphs in Figure 2 in Zhang et al. (2020). Throughout this paper, we do not restrict ourselves from intervening on the variables of interest in the counterfactual query.

## 3 Counterfactual with different interventions

In section 2.1, we mentioned soft interventions, where the original conditional distributions of the intervened variables are replaced with new ones without fully eliminating the causal effect of the parents. This operation is also known as a mechanism change (Tian & Pearl (2013)). It presents in many settings a more realistic model than hard or perfect interventions, where variables are forced to a fixed value. Karimi et al. (2021) and Crupi et al. (2021) perform soft interventions (particularly an additive intervention) to generate counterfactual explanation and recommendation in the context of algorithmic recourse.

**Example 1** (adapted from Example 6.18 in Peters et al. (2017)). *Consider the following SCM:*

$$X := \epsilon_X + 1$$
$$Y := X^2 + \epsilon_Y$$
$$Z := 2Y + X + \epsilon_Z$$

*with $\epsilon_X, \epsilon_Y, \epsilon_Z \sim Uniform(\{-5, -4, ..., 4, 5\})$ idenpendently. Now, assume that we observe $(X, Y, Z) = (1, 2, 4)$ and we are interested in the counterfactual query (a): what would have been $Z$, had $Y$ been 5? Now we pose the question as follows:*

*To answer the counterfactual query, do we need to know the state of the $\epsilon_X$?*

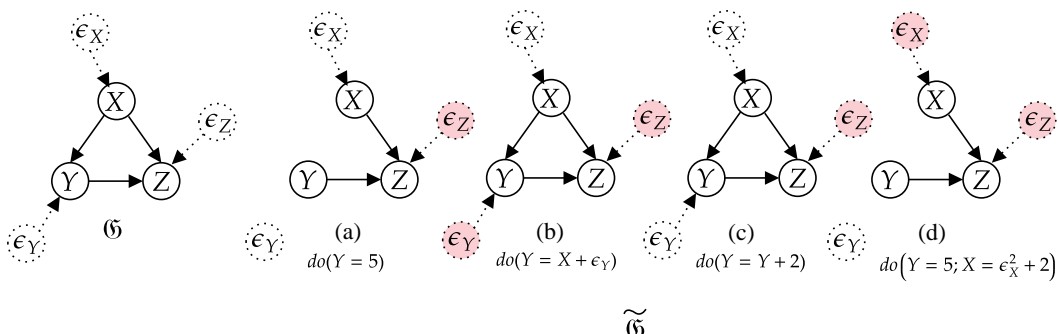

Figure 1: Left-most directed acyclic graph (dag) $\mathfrak{G}$ is the causal graph induced by the SCM in example 1. The rest four are causal graphs induced by counterfactual SCMs for queries in (a),(b),(c) & (d). Noises that must be abducted are filled in pink.

*Note that, given observation $(X, Y, Z) = (1, 2, 4)$, inferring $\epsilon_Z = -1$ is sufficient to answer $Z_{Y \leftarrow 5}(\epsilon) = 10$. Furthermore, We do not even need to know the structural equations of $X$ and $Y$. However, the scenario would be a bit different if we change the counterfactual question (b): What would have been $Z$, had $Y$ followed $Y := X + \epsilon_Y$? In this case, given observation $(X, Y, Z) = (1, 2, 4)$, we need to infer $\epsilon_Z = -1$ and $\epsilon_Y = 1$ to answer that $Z$ would have been 4, had $Y$ followed the structural equation $Y := X + \epsilon_Y$. Further, for computing $Z_{Y \leftarrow Y+2}(\epsilon) = 8$ (c), we do not even need to infer $\epsilon_Y$. Only $\epsilon_Z$ suffices. Here, an interesting observation to make is that the dag $\widetilde{\mathfrak{G}}$ of the manipulated SCM $\widetilde{\mathfrak{C}}$ remains the same as $\mathfrak{G}$ for the counterfactual quarries in (b) and (c) (figure 1). To illustrate more, consider the counterfactual question (d): What would*

*have been $Z$, had $Y$ been 5 and $X$ followed $X := \epsilon_X^2 + 2$? It is sufficient to infer $(\epsilon_X, \epsilon_Z) = (0, -1)$ to answer (d). Had $Y$ been 5 and $X$ followed $X := \epsilon_X^2 + 2$, $Z$ would have been 11.*

The above example motivates us to define semi-hard\semi-soft intervention, an intermediate scenario where we technically do not force a constant value but disregard the interventions on the ancestor variables (of the intervened variable) when we intervene. Semi-hard\semi-soft intervention is defined as taking a unique functional form and, as a result, it is not required to know intervened variable's parents and corresponding noise variable for computing the value of the intervention if we are given the observed value.

**Definition 1** (semi-soft\semi-hard intervention). *An intervention on $X_t$ of the form $X_t \leftarrow \tilde{f}(\mathbf{Pa}_t, \epsilon_t) = h(f(\mathbf{Pa}_t, \epsilon_t))$, where $h$ is any arbitrary function, is called semi-soft\semi-hard intervention.*

$\mathbf{Pa}_t$ emphasizes the fact that we disregard any intervention on ancestors of $X_t$. If we consider the intervention on ancestors, we would have written it with $\widetilde{\mathbf{Pa}_t}$, which coincides with soft intervention. A concrete example is given in Appendix B.1. An typical additive interventions is an example of a semi-soft intervention ($h(f) = f + c$, where $c$ is a constant). Hard interventions are also a special case of semi-soft interventions ($h(f) = c$) but in this article, we strictly differentiate between a hard intervention, a soft intervention and a semi-soft intervention. One may argue that since we are denying interventional changes on ancestors when we are intervening, we could disconnect a semi-hard intervened variable from its parents in the graph induced by interventions. We resort to this argument for the rest of the paper.

## 4 Notations and problem setup

A path in $\mathfrak{G}$ is a sequence of (at least two) distinct vertices $X_{i_1}, ..., X_{i_m}$, such that there is an edge between $X_{i_k}$ and $X_{i_{k+1}}$ for all $k = 1, ..., (m-1)$. If $X_{i_k} \to X_{i_{k+1}}$ for all $k$, we speak of a directed path from $X_{i_1}$ to $X_{i_m}$, denoted as $\mathcal{P}_{i_1 \to i_m}$. We will use the following standard kinship relations for sets of vertices in a directed acyclic graph $\mathfrak{G}$:

$\mathbf{De}_i^{\mathfrak{G}} = \{X_j : \exists \text{ a directed path from } X_i \text{ to } X_j \text{ in } \mathfrak{G}\}$
$\mathbf{De}_A^{\mathfrak{G}} = \{X_j : \exists \text{ a directed path to } X_j \text{ from } X_i \text{ in } \mathfrak{G, for any } i \in A\}$
$\mathbf{An}_i^{\mathfrak{G}} = \{X_j : \exists \text{ a directed path from } X_j \text{ to } X_i \text{ in } \mathfrak{G}\}$
$\mathbf{An}_A^{\mathfrak{G}} = \{X_j : \exists \text{ a directed path from } X_j \text{ to } X_i \text{ in } \mathfrak{G, for any } i \in A\}$

Given an index set $\mathcal{C} \subseteq \{1, 2, ...p\}$, $\mathbf{X}_{\mathcal{C}}$ denotes the random vector $(X_i)_{i \in \mathcal{C}}$ and $\mathbf{X}_{-\mathcal{C}} = (X_i)_{i \notin \mathcal{C}}$. Let us formally state the problem we want to address. Assume $\mathfrak{C} := (\mathcal{S}, \mathbb{P}(\epsilon))$ be a structural causal model. The graph of $\mathfrak{C}$ is $\mathfrak{G}$. For ensuring identifiability, we assume that $\mathfrak{C}$ satisfies four standard assumptions: the Markov property, causally sufficiency (i.e., no hidden confounders), causal minimality, and causal faithfulness (Peters et al. (2017)). Assume $A_H$ to be the index set of random variables on which we perform hard interventions in the action stage. Similarly, $\{X_i : i \in A_S\}$ and $\{X_i : i \in A_T\}$ be the set of random variables on which we act soft interventions and semi-hard\semi-soft interventions, respectively. $A = A_S \cup A_H \cup A_T$ is the index set of intervened variables. Let the counterfactual query we want to answer be $\mathcal{Q}$:

**What would $\mathbf{X}_{\mathcal{C}}$ have been if $\mathbf{X}_{A_H}$ were $\mathbf{x}_{A_H}$ and for each $i \in A_S \cup A_T$, mechanism $f_i$ was changed to $\tilde{f}_i$, given that we observe $\mathbf{X} = \mathbf{x}^{obs}$?**

For the sake of simplicity, we denote the intervention

$$do\left(X_j = x_j \text{ for } j \in A_H; X_j = \tilde{f}_j(\mathbf{Pa}_j, \epsilon_j) \text{ for } j \in A_S \cup A_T\right)$$

as $do(\mathcal{A} \leftarrow \mathfrak{a})$. $\tilde{\mathfrak{G}}$ be the graph of counterfactual SCM $\tilde{\mathfrak{C}}$, modified by intervention $do(\mathcal{A} \leftarrow \mathfrak{a})$. For $i \in \mathcal{C}$, let $X_{i_{\mathcal{A} \leftarrow \mathfrak{a}}}$ denotes an answer to the counterfactual query $\mathcal{Q}$. Set of all directed paths from $\mathbf{X}_A$ to $\mathbf{X}_C$ in $\mathfrak{G}$ is defined as $\mathcal{P}_{\mathfrak{G}}(\mathbf{X}_A \to \mathbf{X}_{\mathcal{C}}) = \{\mathcal{P}_{i \to j} \text{ is in } \mathfrak{G} : i \in A, j \in \mathcal{C}\}$.

## 5 Noises that are essential to $\mathcal{Q}$

**Observation 1.** *If we intervene on $X_j$, following the causal flow in the DAG $\mathfrak{G}$, only $X_j$ and the descendants of $X_j$, $\mathbf{De}_j^{\mathfrak{G}}$ will get affected[3].*

**Theorem 1.** $X_{i\,\mathcal{A}\leftarrow\mathfrak{a}} = x_i^{obs}$ *almost surely, for $i \in \mathcal{C} \setminus \{k : X_k \in \mathbf{De}_A^{\mathfrak{G}} \cup \mathbf{X}_A\}$.*

*Proof.* Consider the subgraph $\mathcal{G}$ of $\mathfrak{G}$, obtained by deleting the vertices in $\{X_k : X_k \in \mathbf{De}_A^{\mathfrak{G}}\} \cup \mathbf{X}_A$. $\tilde{\mathfrak{G}}$ be the graph induced by the SCM $\tilde{\mathfrak{C}}$, modified by the intervention $do(\mathcal{A} \leftarrow \mathfrak{a})$. By observation 1, for any $i \in \{k : X_k \in \mathcal{G}\}$, the triplet $(f_i, \mathbf{Pa}_i^{\mathcal{G}}, \mathbf{An}_i^{\mathcal{G}})_{\mathfrak{C}}$ is same as $(f_i, \mathbf{Pa}_i^{\tilde{\mathfrak{G}}}, \mathbf{An}_i^{\tilde{\mathfrak{G}}})_{\tilde{\mathfrak{C}}}$, where $(f_i, \mathbf{Pa}_i^{\mathcal{G}}, \mathbf{An}_i^{\mathcal{G}})_{\mathfrak{C}}$ denotes the triplet of the structural assignment $f_i$ in the SCM $\mathfrak{C}$, parents and ancestors of $X_i$ in a subgraph $\mathcal{G}$ of $\mathfrak{G}$, respectively. Let $\boldsymbol{\epsilon} = \epsilon$ be one of the situations that leads to the observation $\mathbf{X} = \mathbf{x}^{obs}$, in particular $X_i = x_i^{obs}$. Then, following a topological order in $\mathfrak{G}$,

$$X_i(\epsilon) = x_i^{obs} = f_i(\mathbf{pa}_i, \epsilon_i) = X_{i\,\mathcal{A}\leftarrow\mathfrak{a}}(\epsilon), \quad \forall i \in \{k : X_k \in \mathcal{G}\}.$$

Hence,

$$\{\epsilon : X_i(\epsilon) = x_i^{obs}\} = \{\epsilon' : X_{i\,A\leftarrow a}(\epsilon') = x_i^{obs}\}, \quad \forall i \in \{k : X_k \in \mathcal{G}\}.$$

Using (4) and (5), we get

$$\mathbb{P}(X_{i\,\mathcal{A}\leftarrow\mathfrak{a}} = x_i^{obs} | \mathbf{X} = \mathbf{x}^{obs}) = 1, \quad \forall i \in \{k : X_k \in \mathcal{G}\}.$$

We get the desired result as $\mathcal{C} \setminus \{k : X_k \in \mathbf{De}_A^{\mathfrak{G}} \cup \mathbf{X}_A\} \subseteq \{k : X_k \in \mathcal{G}\}$ .

$\square$

As an immediate consequence, we do not need to infer noises attached to the variables outside the action set $\mathbf{X}_A$ and its descendants $\mathbf{De}_A^{\mathfrak{G}}$, as we are about to modify the SCM $\mathfrak{C}$ by acting on variables in $A$. For example, in the causal graph of figure 2a, if we intervene (hypothetically, in theory) on 'gender', a counterfactual answer about 'age' will not be a diversion from what we observe, which is pretty much intuitive from the causal graph and indeed 'causal' in nature. On the other hand, we are interested in counterfactual queries about $\mathbf{X}_{\mathcal{C}}$. We do not need to oversee all the variables in $\mathbf{De}_A^{\mathfrak{G}} \cup \mathbf{X}_A$.

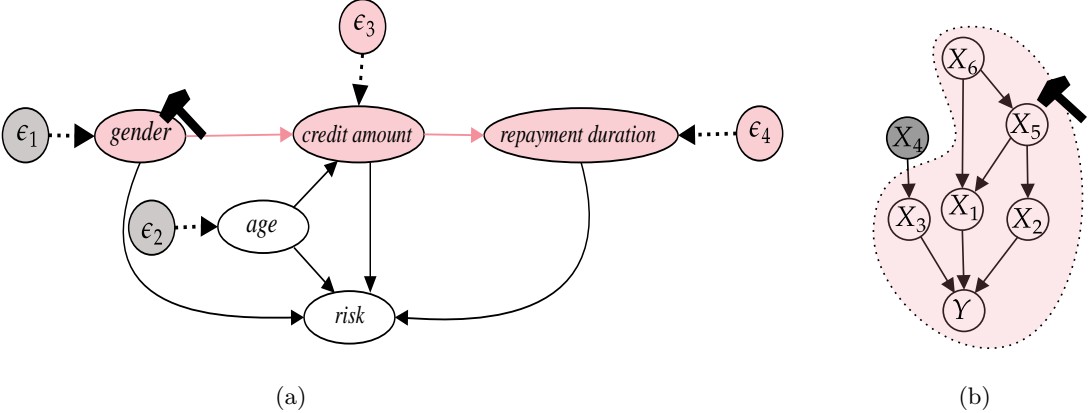

(a)                                                                 (b)

Figure 2: (a) Causal graph for the German credit dataset. (b) Causal graph of the synthetic dataset.

**Theorem 2.** *Assume that $X_j$ has not been intervened. Then counterfactual prediction on $X_j$ may differ from its observed value $x_j^{obs}$ iff at least an ancestor of $X_j$ has been intervened.*

---

[3]By 'get affected', we mean a possibility of distributional change.

*Proof.* If we intervene on an ancestor of $X_j$, from observation 1, counterfactual prediction on $X_j$ may differ from its observed value $x_j^{obs}$. For the only if part, assume none of the ancestors of $X_j$ has been intervened. Let $I$ be the index set of intervened variables, then $X_j \notin \mathbf{De}_I^{\mathfrak{G}}$. Moreover, as $X_j$ hasn't been intervened on, by theorem 1, the counterfactual value of $X_j$ remains the same as its observed value, contradicting the hypothesis.

$\square$

Theorem 2 says we need to worry about noises attached to variables in $\mathbf{An}_{\mathcal{C}}^{\mathfrak{G}} \cup \mathbf{X}_{\mathcal{C}}$ only, as we are interested in a counterfactual query about $\mathbf{X}_{\mathcal{C}}$. For example, if we are concerned about only 'repayment duration' in the causal graph of figure 2a, we need to take care of its ancestors' exogenous noise. Furthermore, theorem 1 and theorem 2 allow us to constrain the search space to all the exogenous noises corresponding to the variables lying on a directed path from $\mathbf{X}_A$ to $\mathbf{X}_{\mathcal{C}}$ in $\mathfrak{G}$. Continuing with the example of figure 2a, if we are interested in 'repayment duration' and we are intervening on 'gender' in the action step, we only need to infer $\epsilon_3$ and $\epsilon_4$ as they are attached to the variables that lie on the directed path (coloured in pink) from 'gender' to 'repayment duration'. Then why do we exclude $\epsilon_1$ from abduction?

**Theorem 3.** $X_{j\,X_j \leftarrow x_j'} = x_j'$.

*Proof.* Immediate from property 2 (Effectiveness) in Pearl (2009). $\square$

Effectiveness property releases us from inferring $\epsilon_{A_H}$. By definition of semi-soft intervention, we do not need to infer $\epsilon_{A_T}$. As the hard interventions and the semi-soft\semi-hard disconnect parents from the intervened variables, we further filter out exogenous disturbances by looking at $\tilde{\mathfrak{G}}$ instead of $\mathfrak{G}$.

**Theorem 4.** $\mathbf{X}_{\mathcal{C}\,\mathcal{A}\leftarrow\mathfrak{a}}(\epsilon) = \mathbf{X}_{\mathcal{C}\,\mathcal{A}\leftarrow\mathfrak{a};do_{\mathcal{A}}^*}(\epsilon)$, where $do_{\mathcal{A}}^* = do\left(X_i = x_i^{obs} \text{ for } X_i \in \mathbf{An}_{\mathcal{C}}^{\tilde{\mathfrak{G}}} \setminus \{\mathbf{De}_A^{\tilde{\mathfrak{G}}} \cup \mathbf{X}_A\}\right)$.

*Proof.* Immediate from theorem 1 and property 1 (Composition) in Pearl (2009) . $\square$

Theorem 4 allows us to intervene on the variables outside $\mathbf{De}_A^{\tilde{\mathfrak{G}}} \cup \mathbf{X}_A$ with their observed values. This intervention $do_{\mathcal{A}}^*$ depends on the intervention $do(\mathcal{A} = \mathfrak{a})$. Theorem 4 also guarantees that $do_{\mathcal{A}}^*$ doesn't change unit-level counterfactuals. We discuss this idea of intervention with the observation in Appendix B.2 in more detail. The set of noises that lie on a path from $\mathbf{X}_A$ to $\mathbf{X}_{\mathcal{C}}$ in $\tilde{\mathfrak{G}}$, i.e., $\{\epsilon_i\}_{i\in\mathfrak{p}}$, where $\mathfrak{p}$ is the index set

$$\mathfrak{p} = \{i : X_i \text{ lies on a path } \mathcal{P} \text{ such that } \mathcal{P} \in \mathcal{P}_{\tilde{\mathfrak{G}}}(X_A \to X_{\mathcal{C}}) \text{ and } \epsilon_i \text{ is exogenous parent of } X_i \text{ in } \tilde{\mathfrak{G}}\},$$

is sufficient to answer $\mathcal{Q}$. We next define the sufficient and the essential set of exogenous noises to answer $\mathcal{Q}$ and then we prove that $\{\epsilon_i\}_{i\in\mathfrak{p}}$ is essential.

**Definition 2** (sufficient and essential set of exogenous noises). *$\bar{\epsilon} \subseteq \{\epsilon_i\}_{i=1}^n$ is said to be sufficient to a counterfactual query $\mathcal{Q}$ if $\mathcal{Q}$ can be answered (or computed) by inferring $\bar{\epsilon}$ only, using the three-step (abduction, action and prediction) algorithm (as described in subsection 2.2). If the sufficient set $\bar{\epsilon}$ is minimal, i.e., any proper subset of $\bar{\epsilon}$ is not sufficient, then $\bar{\epsilon}$ is called essential.*

**Theorem 5.** $\{\epsilon_i\}_{i\in\mathfrak{p}}$ *is essential to $\mathcal{Q}$.*

*Proof.* Assume that we do not infer $\epsilon_j$, $j \in \mathfrak{p}$. If $j \in \mathfrak{p} \cap A_S$, i.e., $X_j$ has been soft intervened on, then the prediction step on $X_j$ based on $\tilde{\mathfrak{C}}$ isn't possible as it requires to compute $\tilde{f}_j(pa_{j\,\mathcal{A}\to\mathfrak{a}}^{\mathfrak{G}}, \epsilon_j)$ and $\epsilon_j$ is unknown. Similarly, if $j \in \mathfrak{p} \setminus A_S$, i.e., $X_j$ has not been intervened on (but at least one of its ancestors has been intervened), then also the prediction step on $X_j$ is not possible as unknown $\epsilon_j$ creates the bottleneck in computing $f_j(pa_{j\,\mathcal{A}\to\mathfrak{a}}^{\tilde{\mathfrak{G}}}, \epsilon_j)$. Note that, $X_j \in \mathbf{An}_{\mathcal{C}}^{\mathfrak{G}} \cup \mathbf{X}_{\mathcal{C}}$. If $j \in \mathcal{C}$, since counterfactual prediction about $X_j$ isn't possible, we are done. If $X_j \in \mathbf{An}_{\mathcal{C}}^{\mathfrak{G}}$ , i.e., $X_j$ is ancestor of at least one variable $X_i$ for $i \in \mathcal{C}$, following the recursiveness of SCM, counterfactual prediction about $X_i, i \in \mathcal{C}$ is not possible. $\square$

We devise the following four-step procedure (adding one more to Pearl (2009)) for computing a counterfactual query $\mathcal{Q}$ in the SCM framework:

1. **Pre-abduction:** Identify the acting interventions, $do(\mathcal{A} \leftarrow \mathfrak{a})$. Identify the essential set of exogenous noises $\{\epsilon_i\}_{i \in \mathfrak{p}}$ for $\mathcal{Q}$.

2. **Abduction:** Predict the essential exogenous noises, $\epsilon_i$'s from the observations $\mathbf{x}^{obs}$, i.e., infer $\mathbb{P}(\pi_{\mathcal{A}}(\boldsymbol{\epsilon})|\mathbf{X} = \mathbf{x}^{obs})$, where $\pi_{\mathcal{A}}$ is a projection operator depending on $do(\mathcal{A} \leftarrow \mathfrak{a})$, maps $\boldsymbol{\epsilon}$ to $\boldsymbol{\epsilon}_{\mathfrak{p}}$.

3. **Action:** Perform the desired interventions $do(\mathcal{A} \leftarrow \mathfrak{a}), do_{\mathcal{A}}^*$.

4. **Prediction:** Compute the quantities of interest in $\mathcal{Q}$.

What pre-abduction says is that - we know the interventions we will perform. Hence a priori, we know causal graph modified by the interventions. So it suggests exploiting this a priori knowledge for noise abduction since we ultimately perform prediction following these interventions and modified causal graph. This exploitation reduces the number of noises needed to abduct from the number of nodes in $\mathfrak{G}$ to the total number of nodes in all directed paths from $\mathbf{X}_A$ to $\mathbf{X}_{\mathcal{C}}$ in $\tilde{\mathfrak{G}}$. This is quite effective in causal graph $\mathfrak{G}$ consists of a moderate or large number of nodes (variables).

# 6 Experiments

## 6.1 Case study 1: synthetic dataset

For the synthetic setting, we generate data following the model in figure 2b, where we assume

$$
\begin{aligned}
&X_6 = \epsilon_6 - 1, && X_5 = 3X_6 + \epsilon_5 - 1, && X_4 = 2\epsilon_4 + 1, \\
&X_3 = -3X_4 + \epsilon_3 - 3, && X_2 = X_5 - \epsilon_2, && X_1 = X_6 - X_5 + 3\epsilon_1, \\
&&& Y = X_1 + 2X_2 - 3X_3 + \epsilon_Y,
\end{aligned}
$$

and $\epsilon_Y, \epsilon_i \sim \mathcal{N}(0, 1)$ independently, for $i = 1, 2..., 6$. We generate 20000 data points from the SCM. This simple dataset allows for a comparison of generated counterfactuals in a controlled and measurable environment. We consider two models to answer: "What would have happened to $Y$, if $X_5$ was different than what we observed: $\mathbf{X} = \mathbf{x}^{obs}, Y = y^{obs}$. The full model infers all the exogenous noises, whereas the partial model only infers $\epsilon_1, \epsilon_2$, and $\epsilon_Y$ (following pre-abduction). We use this setting to study the importance of noise identification for abduction.

We use affine coupling flows (Dinh et al., 2017) for $X_4$ and $X_6$ and conditional affine coupling transform for other dependent variables. In the full model, seven flows are implemented - two linear flows and five conditional flows. In the partial model, only three conditional flows are used for $X_1, X_2$, and $Y$. respectively. We model base densities with standard gaussian.

We use the Pyro (Bingham et al., 2019) probabilistic programming language (PPL) framework for the implementation of the flow-based SCM. A PPL is a programming language in which probabilistic models are specified and inference for these models is performed automatically with terms corresponding to sampling and conditioning. Pyro is a PPL based on PyTorch (Paszke et al., 2019). For a detailed overview of PPLs, see van de Meent et al. (2018). Adam (Kingma & Ba, 2015) with batch-size 128, an initial learning rate of $10^{-3}$ is used for optimization purposes. Both models are trained for 1000 epochs using 12th Gen Intel(R) Core(TM) i9-12900KF CPU.

Figure 3a shows the counterfactual distribution $\mathbb{P}(Y_{X_5 \leftarrow 5}|\mathbf{X} = \mathbf{x}^{obs}, Y = y^{obs})$ estimated by the full and partial models (for one particular seed) along with the true counterfactual distribution. We quantitatively compare the associative capabilities of both models by log-likelihoods (validation) as shown in the table 1. Figures depicting the goodness of noise estimation and sampling capabilities of both models are provided in

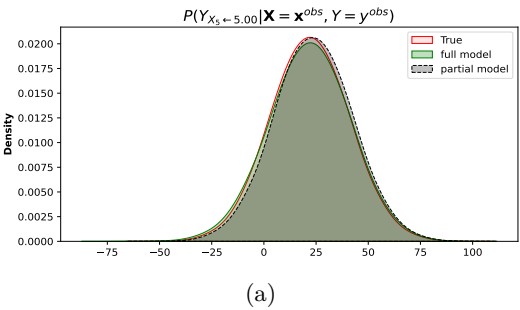 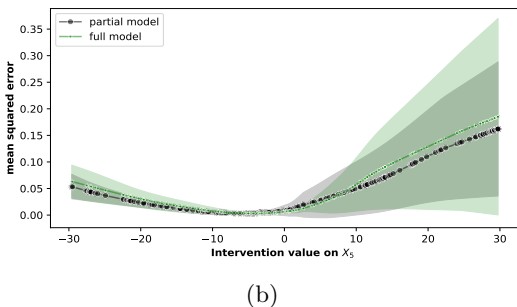

(a)          (b)

Figure 3: (a) The red curve is the kernel density estimate (KDE) plot of the true counterfactual distribution. The solid green and black dashed lines are the KDE plots of the distributions estimated (for one particular seed) by the full and partial models. (b) Average (over ten different seeds) mean squared errors in estimating counterfactual values of $Y$. The $x$-axis represents the values we intervene on $X_5$ Black circles are average errors in the partial model. Green dots are average errors in the full model.

Appendix C. We run the same experiment for ten different seeds. We intervene $X_5$ with 200 different values[4] uniformly sampled from -30 to 30. Average (over ten different seeds) mean squared errors in counterfactual estimation (on seen datapoints[5]) for each model for the 200 different intervention values have been depicted in figure 3b. We report the average time to train 1000 epochs for both models in table 1.

Table 1: Best validation log-likelihood and average training time for the full and partial model.

| Model | Log-likelihood | | | | | | | Training time |
| --- | --- | --- | --- | --- | --- | --- | --- | --- |
| | $X_6$ | $X_5$ | $X_4$ | $X_3$ | $X_2$ | $X_1$ | $Y$ | (in min.) |
| Partial | — | — | — | — | **-1.4160** | **-2.5050** | **-1.4415** | **6.46 ± 0.033** |
| Full | -1.4198 | -1.4166 | -2.1126 | -1.4229 | -1.4163 | -2.5050 | -1.4418 | 11.22 ± 0.081 |

Next, we experiment with training time progression (for 100 epochs) with different batch and sample sizes for both partial and full models. Samples are generated from the same SCM. We run both models for ten different seeds. Figure 4 depictures the average (over ten different seeds) training time ratios (partial model/full model) with different sample sizes and four different batch sizes.

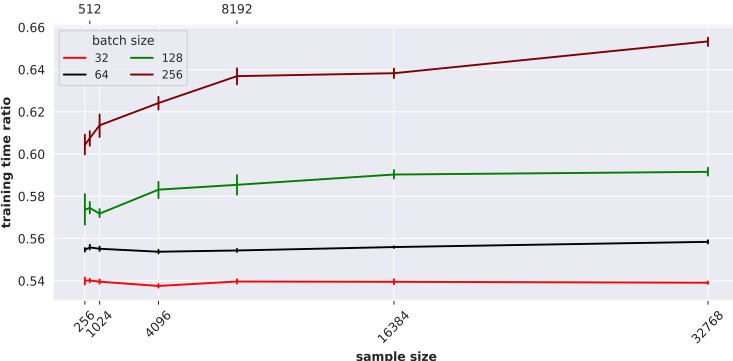

Figure 4: Training time ratio (partial model/full model) vs. sample size for four different batch sizes.

---

[4]These 200 values remain the same across ten different seeds.
[5]By 'seen datapoints', we mean these are the datapoints used in training and validation. MSE in estimation of counterfactuals on unseen data points( test MSE) is given in Appendix C

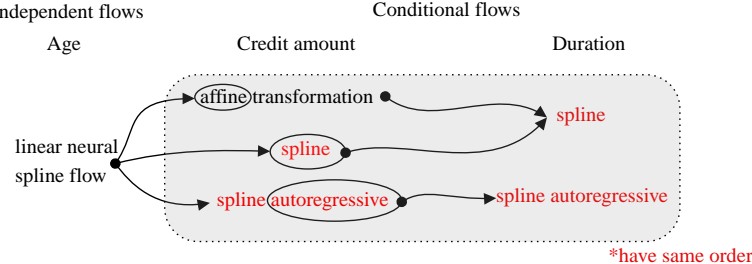

Figure 5: Combinations of flows used in the experiment. Flows' combinations are identified by the phrases inside the ellipse. Flows inside the light grey rectangle are used in the partial model, i.e., we don't model any flow for age in the case of the partial model. We have the same order (i.e., either linear or quadratic) for the flows in red.

## 6.2 Case study 2: german credit dataset

As a real-world setting, we consider a subset of the features in the german credit dataset. This subset includes gender ($X_1$), age ($X_2$), credit amount ($X_3$), and repayment duration ($X_4$). In figure 2a, we see an example of DAG representing the causal relationships (Karimi et al. (2021)) in the german credit dataset (Dua & Graff (2017)). We do not consider the risk variable in our experiment. We are interested in studying the counterfactual query: Had the person been male instead of female (or female instead of male), would the person has been offered more (or less) credit amount for a larger ( or shorter) duration?

First, flow-based SCM is trained using the observed data. Next, the states of exogenous noises are inferred with the estimated structural assignments that are invertible (abduction step). Then we intervene upon the sex by replacing the sex variable with a specific value 'male' or 'female'; this is denoted by do(sex = male) or do(sex=female). We use the modified flow-based SCM to compute counterfactual quantities. Similar to the synthetic data experiment, we consider two models. The full model infers all the exogenous noise variables except $\epsilon_1$ since we model the mechanisms of gender\sex($X_1$) as $x_1 = f_1(\epsilon_1) = \epsilon_1$. Age $X_2$, Credit amount $X_3$ and repayment duration $X_4$ are modelled as

$$x_2 = f_2(\epsilon_2) = (\text{Spline}_\theta \circ \text{AffineNormalisation} \circ \exp)(\epsilon_2),$$
$$x_3 = f_3(\epsilon_3; x_1, x_2) = (\text{ConditionalTransform}_\theta([x_1, x_2]) \circ \text{AffineNormalisation} \circ \exp)(\epsilon_3),$$
$$x_4 = f_4(\epsilon_4; x_3) = (\text{ConditionalTransform}_\theta([x_3]) \circ \text{AffineNormalisation} \circ \exp)(\epsilon_4).$$

The modules highlighted by $\theta$ are parameterized using neural networks. We use a categorical distribution for sex ($X_1$) and directly learn the binary probability of sex ($X_1$). The densities of exogenous noises (except $\epsilon_1$) are standard gaussians. For other structural assignments, we use real-valued normalizing flows. A linear flow and two conditional flows (conditioned on activations of a fully-connected network, one takes age and sex as input for credit amount and another takes credit amount as input for the duration) are used as structural assignments for age, credit amount, and duration features, respectively. We constrain age ($X_1$), credit amount ($X_3$), and repayment duration ($X_4$) variables with lower bound (exponential transform) and rescale them using a fixed affine transform for normalization. The partial model infers only $\epsilon_3$ and $\epsilon_4$ as suggested by pre-abduction (described in section 5). We model flows for credit amount ($X_4$) and repayment duration ($X_3$) similar to the full model. However, we do not model a flow for the age variable. Combinations of flows used in the experiment are depicted in figure 5.

Spline$_\theta$ transformation stands for the linear neural spline flows (Dolatabadi et al. (2020).) ConditionalTransform$_\theta(\cdot)$ can be conditional affine or conditional spline transform. We use linear (Dolatabadi et al. (2020)) and quadratic (Durkan et al. (2019)) order, autoregressive and linear neural spline flows for the conditional spline transform. These are more expressive in comparison to the affine flows. Taking $\cdot$ as input, a context neural network estimates the transformation parameters of the ConditionalTransform$_\theta(\cdot)$. We implement the context networks as fully-connected networks for spline and affine flows. Adam (Kingma

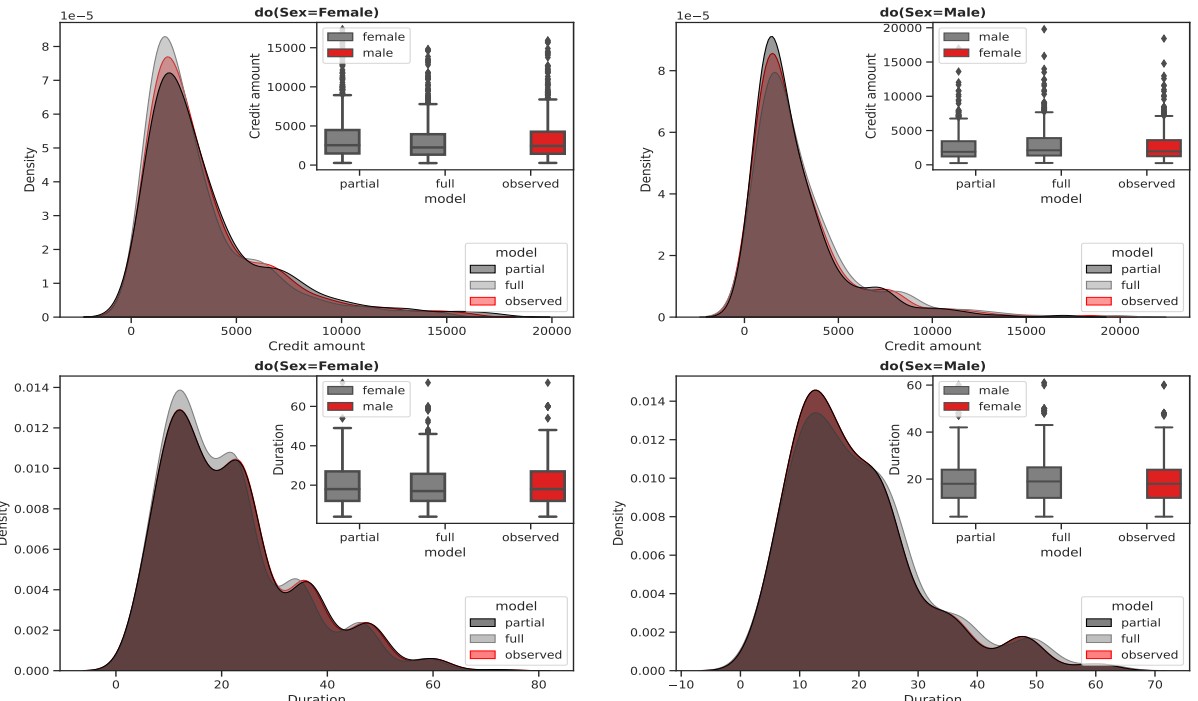

Figure 6: On the left, KDE plots of the observed distributions $P(\text{Credit amount}|\text{Sex} = \text{male})$ and $P(\text{Repayment duration}|\text{Sex} = \text{male})$ are given in red. Counterfactual distributions $P(\text{Credit amount}_{do(\text{Sex=female})}|\text{Sex} = \text{male})$, $P(\text{Repayment duration}_{do(\text{Sex=female})}|\text{Sex} = \text{male})$ estimated by full and partial models are presented in gray and black, respectively. On the right, KDE plots of the observed distributions $P(\text{Credit amount}|\text{Sex} = \text{female})$ and $P(\text{Repayment duration}|\text{Sex} = \text{female})$ are given in red. Counterfactual distributions, $P(\text{Credit amount}_{do(\text{Sex=male})}|\text{Sex} = \text{female})$ and $P(\text{Repayment duration}_{do(\text{Sex=male})}|\text{Sex} = \text{female})$, estimated by full and partial models, are presented in gray and black, respectively. The upper panel is for distributions related to credit amounts. The lower panel is for distributions related to payment duration. Box plots at the right-hand corner of each subplot are self-explanatory.

& Ba, 2015) with a batch-size of 64, an initial learning rate of $3 \times 10^{-4}$, and weight decay of $10^{-4}$ are used in training. We use a staircase learning rate schedule with decay milestones at 50% and 75% of the training duration. All instances of both models are trained for 500 epochs using NVIDIA RTX A5000 GPU. Training times are reported in Appendix D.

Figure 6 depicts how observed distributions of credit amounts and repayment duration would have changed to the corresponding counterfactual distributions if we hypothetically set the gender of the loanees different from what is reported. While we present the result of counterfactual estimation via 'affine' flow combinations of linear order in figure 6, the results of other flow combinations are in Appendix D. We also quantitatively compare the associative capabilities of all instances of both models by log-likelihoods (validation) as given in Appendix D.

## 7 Discussion

This paper tackles the problem of identifying exogenous noises that must be abducted for counterfactual inference. We demonstrate that explicitly identifying noises is an important task for counterfactual inference as we empirically show that identifying noise variables can reduce the computational load of counterfactual inference without compromising in performance. Identifying exogenous noise variables for answering

a counterfactual query also reduces the burden of modelling too many normalizing flows. Our work makes Pawlowski et al. (2020)'s framework applicable to partially specified causal graphs in the setting where we observe all variables that lie in a directed path from $X_A$ to $X_C$ along with their parents. The causal relations among these variables are needed to be fully specified. For example, consider the causal graph in figure 2b. If we are interested in $Y_{X_5 \leftarrow x_5'}(\epsilon)$, it does not matter whether $X_4$ is observed or not. Sub-graph inside the pink region will suffice. Note that we haven't really used $X_4$ in the partial model of synthetic data experiment as we conditioned on $X_3$ for answering $Y_{X_5 \leftarrow x_5'}(\epsilon)$.

Though our work is heavily inspired by Pawlowski et al. (2020)'s framework, it is very general to apply to other frameworks for generating counterfactuals. Our work does come with limitations to be investigated further. For example, we do not study the scenario when a hidden (unobserved) variable lies in a path from the intervened variable to the variable we are interested in. We fundamentally do not restrict ourselves from intervening on the variables. In scenarios where we can not intervene on a variable fundamentally, i.e., when we try to answer the counterfactual queries from observed or a combination of observed and experimental data only, identification of counterfactual questions itself is the first priority. It would be interesting to investigate the roles of the noises in such settings. Another limitation is that reducing the noise abduction set might restrict the generative power of the model[6].

A noted limitation in counterfactual inference is that counterfactuals are usually unverifiable for real datasets. Evaluation is not possible except in a few constrained settings, as true counterfactuals are never usually noticed. Counterfactual speculation is what some variable would be in a parallel universe where all but the intervened variables and their descendants were the same. However, the machinery of counterfactual inference provides scientists with better schemes for controlling the known confounders. As a result, the SCM framework is widely applicable to enhance trust and the performance of ML\AI systems.

### Acknowledgments

This research is partially supported by Science and Engineering Research Board (SERB), Dept. of Science and Technology (DST), Govt. of India through Grant File No. SPR/2020/000495.

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

## A    Normalizing Flows

Normalizing flows learn complex probability distributions of real data using a sequence of diffeomorphic transformations from simpler base distributions with the same dimensionality. For an observed variable $X_i$, diffeomorphic transformations $g_1^i, g_2^i, ..., g_{i_K}^i$ and base variable $\epsilon_i \sim \mathbb{P}(\epsilon_i)$ such that $X_i = (g_{i_K}^i \circ ... \circ g_2^i \circ g_1^i)(\epsilon_i) = f_i(\epsilon_i)$, the target density $\mathbb{P}(x_i)$ can be calculated as

$$\mathbb{P}(x_i) = \mathbb{P}(\epsilon_i) \mid \det \nabla f_i(\epsilon_i) \mid^{-1},$$

evaluated at $\epsilon_i = f_i^{-1}(x_i)$. Assuming $g_0^i$ as identity function,

$$\log \mathbb{P}(x_i) = \log \mathbb{P}(\epsilon_i) + \log \mid \det \nabla f_i(\epsilon_i) \mid^{-1}$$

$$= \log \mathbb{P}(\epsilon_i) + \log \mid \det \prod_{j=1}^{i_K} \nabla g_j^i \mid_{(g_{j-1}^i \circ ... \circ g_2^i \circ g_1^i)(\epsilon_i)} \mid^{-1}$$

$$= \log \mathbb{P}(\epsilon_i) - \sum_{j=1}^{i_K} \log \mid \det \nabla g_j^i \mid_{(g_{j-1}^i \circ ... \circ g_2^i \circ g_1^i)(\epsilon_i)} \mid .$$

As the exact log-likelihood of input data becomes tractable, the loss function is simply the negative log-likelihood, and the model explicitly learns the data distribution. Moreover, It is possible to make flows as expressive as needed. In particular, for any pair of well-behaved distributions $\mathbb{P}(x_i)$ and $\mathbb{P}(\epsilon_i)$, there exists a diffeomorphism $f_i$ that can turn $\mathbb{P}(\epsilon_i)$ into $\mathbb{P}(x_i)$ (Papamakarios et al., 2019). Trippe & Turner (2018) has extended normalizing flows to conditional densities by parametrising the transformation as $x_i = f_i(\epsilon_i; \mathbf{pa}_i)$. Note that invertibility is assumed in the first argument.

## B    A deeper look

### B.1    Semi-hard intervention

Consider the following SCM

$$
\begin{aligned}
X &= f_X(\epsilon_X) = \epsilon_X, & \epsilon_X &\sim \mathcal{N}(0,1), \\
Y &= f_Y(X; \epsilon_Y) = X^2 + \epsilon_Y, & \epsilon_Y &\sim \mathcal{N}(0,1), \\
Z &= f_Z(Y, X, \epsilon_Z) = X + Y^2 + \epsilon_z, & \epsilon_Z &\sim \mathcal{N}(0,1).
\end{aligned}
$$

Consider the mechanism change $do\left(X = \tilde{f}_X(\epsilon_X) = \epsilon_X + 1, Y = \tilde{f}_Y(X; \epsilon_Y) = X + \epsilon_Y + 1\right)$. Now the fundamental question is - In the intervention $Y \leftarrow \tilde{f}_Y(X; \epsilon_Y) = X + \epsilon_Y + 1$, which structural equation of $X$ we should consider? $f_X$ or $\tilde{f}_X$? If $\tilde{f}_X$ is taken into account, then it can be interpreted as a standard soft intervention. This mechanism change may be seen as a combination of sequential interventions. If we consider $f_X$, then we are disregarding the intervention on its parent, i.e., $do\left(X = \tilde{f}_X(\epsilon_X) = \epsilon_X + 1\right)$. In general, we disregard interventions on ancestors. We can think of this mechanism change as a combination of simultaneous interventions. To emphasize which structural assignment of $X$ has been taken into account when we intervene on $Y$, we write it as $Y \leftarrow \tilde{f}_Y(\widetilde{Pa_Y}, \epsilon_Y)$ and $Y \leftarrow \tilde{f}_Y(Pa_Y, \epsilon_Y)$ for $\tilde{f}_X$ and $f_X$, respectively.

Semi-hard\semi-soft intervention on a variable $X_t$ is just a soft intervention with $\tilde{f} = h(f(Pa_t, \epsilon_t))$, disregarding interventions on $X_t$'s ancestors. For example, $Y \leftarrow Y^2 + 2$ in the given SCM.

### B.2    What pre-abduction is doing? Intervene with observation?

In an SCM $\mathfrak{C}$ with the graph $\mathfrak{G}$,

$$X_i(\epsilon) = f_i(\mathbf{pa}_i^{\mathfrak{G}}, \epsilon_i),$$

where $X_i(\epsilon)$ expresses $X_i$'s dependency on exogenous noises $\epsilon$ only. The functional form of the $X_i(\epsilon)$ can be obtained by substituting variables with their structural assignments following a reverse topological order in $\mathfrak{G}$ starting from $f_i(\mathbf{pa}_i^{\mathfrak{G}}, \epsilon_i)$. From the computational perspective, the difference between $X_i(\epsilon)$ and $f_i(\mathbf{pa}_i^{\mathfrak{G}}, \epsilon_i)$ is what you need to know for computing $X_i$. Further incorporating a projection map $\epsilon_i = \pi_i(\epsilon)$, we can write it as

$$X_i(\epsilon) = f_i(\mathbf{pa}_i^{\mathfrak{G}}, \epsilon_i) = f_i(\mathbf{pa}_i^{\mathfrak{G}}, \pi_i(\epsilon)).$$

Let $\epsilon$ be a situation that leads to $\mathbf{x}^{obs}$. Note that, $\forall \epsilon' \in \pi_i^{-1}(\epsilon_i)$, the following statement may **not** hold

$$X_i(\epsilon') = f_i(\mathbf{pa}_i^{\mathfrak{G}}, \epsilon_i) = f_i(\mathbf{pa}_i^{\mathfrak{G};obs}, \epsilon_i) = f_i(\mathbf{pa}_i^{\mathfrak{G};obs}, \pi_i(\epsilon')). \tag{6}$$

Why do we even want such a thing to hold true? Because then it does not matter whether we know $\epsilon_j, j \neq i$ or not. This can be best understood with the following example.

**Example 2.** *Consider the following SCM,*

$$X = \epsilon_X^2$$
$$Y = (X - 1)^2 + \epsilon_Y^2$$

*and observation* $(x^{obs}, y^{obs}) = (1, 0)$. *This observation could have arrived in situations* $\epsilon = (\epsilon_X, \epsilon_Y) = \{(1, 0), (-1, 0)\}$. $\pi_Y^{-1}(\{0\}) = \mathbb{R} \times \{0\}$. *Let* $(r, 0) \in \pi_Y^{-1}(\{0\})$ *and* $r^2 \neq 1$.

$$X((r, 0)) = f_Y(x(r), 0) = (r^2 - 1)^2 \neq 0 = f_Y(x = x^{obs}, \pi_Y(r, 0)).$$

Expression in 6 holds $\forall \epsilon' \in \pi_i^{-1}(\epsilon_i)$, if we intervene on $\mathbf{Pa}_i^{\mathfrak{G}}$ with $\mathbf{pa}_i^{\mathfrak{G};obs}$. In particular, this intervention induce the graph $\tilde{\mathfrak{G}}$ and $\forall \epsilon' \in \pi_i^{-1}(\epsilon_i)$,

$$X_i(\epsilon') = f_i(\mathbf{pa}_i^{\tilde{\mathfrak{G}}}, \epsilon_i) = f_i(\mathbf{pa}_i^{\mathfrak{G};obs}, \epsilon_i) = f_i(\mathbf{pa}_i^{\mathfrak{G};obs}, \pi_i(\epsilon')).$$

For any given intervention set $I \subseteq \mathbf{X} \setminus X_i$, $X_i$ can be expressed as

$$X_i(\boldsymbol{\epsilon}) = g_i\Big(I \cap \mathbf{An}_i^{\tilde{\mathfrak{G}}}, \mathbf{B}, \pi_I(\boldsymbol{\epsilon})\Big),$$

where $\mathbf{B} \subseteq \mathbf{An}_i^{\tilde{\mathfrak{G}}} \setminus (\mathbf{De}_I \cup I)$. If $I = \phi$, then $\tilde{\mathfrak{G}} = \mathfrak{G}, g_i = f_i, B = \mathbf{Pa}_i^{\mathfrak{G}}, \pi_I = \pi_i$. Such an expression can be obtained by traversing reverse on the directed paths from $I$ to $X_i$ in $\mathfrak{G}$ and replacing visiting nodes with their structural assignment.

Consider the intervention $I = \mathcal{A}$ with index set $A$. Counterfactual inference with the pre-abduction step comes with an inherent intervention - $do\Big(\mathbf{An}_i \setminus (\mathbf{De}_A \cup \mathbf{X}_A) = (\mathbf{an}_i \setminus (\mathbf{de}_A \cup \mathbf{x}_A))^{obs}\Big)$. For the sake of simplicity, we refer to this as $do_{\mathcal{A}}^*$.

## C  Synthetic data experiment

### C.1  Goodness of abduction (specific to one particular seed)

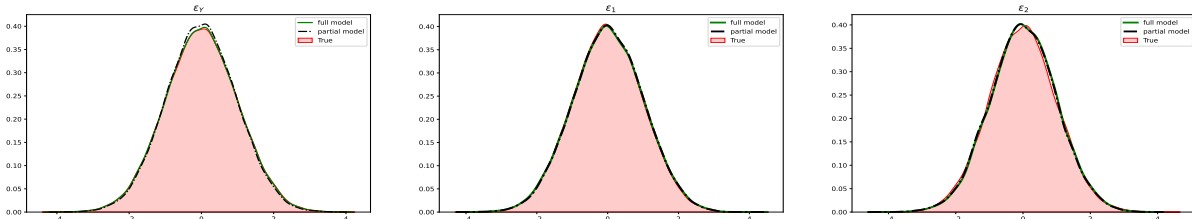

Figure 7: KDE plots of true exogenous noise data ($\epsilon_Y, \epsilon_1, \epsilon_2$) are in red. KDE plots of the exogenous noises estimated by the partial and the full model are in black and green, respectively.

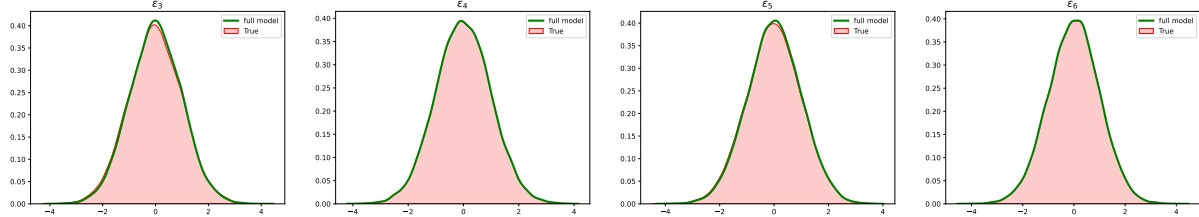

Figure 8: KDE plots of true exogenous noise data ($\epsilon_3, \epsilon_4, \epsilon_5, \epsilon_6$) are in red. KDE plots of the exogenous noises estimated by the full model are in green.

The full model abducts all the noise variables. On the other hand, the partial model abducts only $\epsilon_Y, \epsilon_1$, and $\epsilon_2$. Here, we compare both models in terms of their ability to infer $\epsilon_Y, \epsilon_1$, and $\epsilon_2$ in figure 7. For the sake of completeness, the full model's ability to infer $\epsilon_3, \epsilon_4, \epsilon_5$, and $\epsilon_6$ is given in figure 8.

### C.2  Sampling abilities of the partial and the full model (specific to one particular seed)

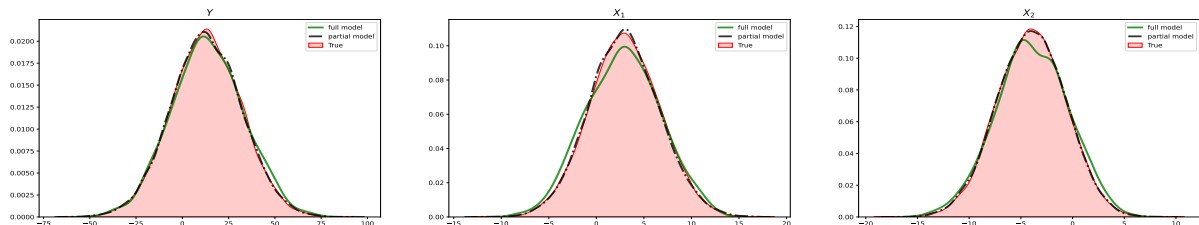

Figure 9: KDE plots of observed data ($Y$, $X_1$, and $X_2$) are in red. KDE plots of the generated samples (1000 points) from the full and the partial model are in black and green, respectively.

As we do not estimate flows for $X_3, X_4, X_5$, and $X_6$ in the partial model, we can not sample for any variable from the SCM using the partial model. Given $X_3, X_4, X_5$, and $X_6$, we can sample only for $X_1, X_2$, and $Y$ using partial model. The full model does not have such limitations. The sampling abilities of both models for $X_1, X_2$ and $Y$ are depicted in figure 9. The sampling ability of full model for $X_3, X_4, X_5$, and $X_6$ is given in figure 10. Note that, if we want to sample from a particular variable from the SCM, we can take care of that variable in the pre-abduction step. For example, if we do not want to lose the sampling ability for $X_4$ from the SCM in figure 2b, we will estimate flows for $X_3$ and $X_4$ in addition to $X_1, X_2$, and $Y$.

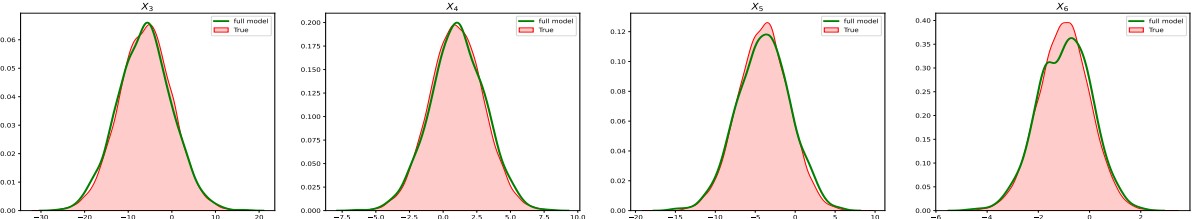

Figure 10: KDE plots of observed data $(X_3, X_4, X_5$ and $X_6)$ are in red. KDE plots of the generated samples (1000 points) from the full model are in green. Partial model isn't able to generate samples for these variables.

## C.3 Average Mean squared error in the estimation of counterfactuals on unseen data

We generate 20000 datapoints from the same SCM for ten different seeds. These datapoints have not been used in training and validation. We try to estimate counterfactuals on these points using the trained model. We intervene $X_5$ with 200 different values[7] uniformly sampled from -30 to 30. Average (over ten different seeds) mean squared errors in counterfactual estimation for each model for the 200 different intervention values have been depicted in figure 11.

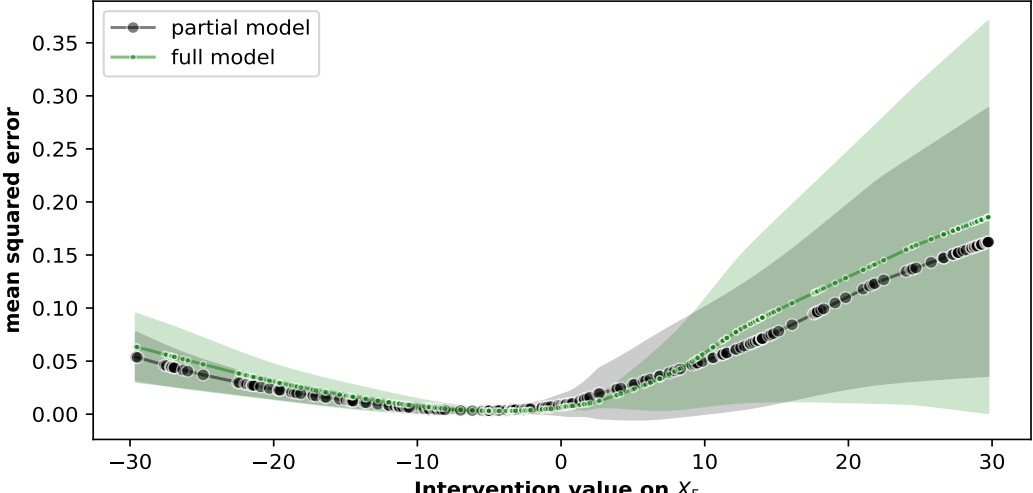

Figure 11: Average (over ten different seeds) mean squared errors in estimating counterfactual values of $Y$. The $x$-axis represents the values we intervene on $X_5$. Black circles are average errors in the partial model. Green dots are average errors in the full model.

---

[7]These 200 values remain the same across ten different seeds.

# D   German credit dataset

Table 2: Best validation -ve log-likelihood and training time for each model.

| Combination of Flows | Flow Order | Model | -ve Log-likelihood | | | | Training time |
| --- | --- | --- | --- | --- | --- | --- | --- |
| | | | Age | Sex | Amount | Duration | (in min) |
| Affine | linear | partial | — | 0.6519 | 9.0625 | 3.5161 | 1.63 |
| | | full | 3.8448 | 0.6519 | 8.9190 | 3.3912 | 1.98 |
| | quadratic | partial | — | 0.6520 | 9.0989 | 3.3936 | 1.58 |
| | | full | 3.8492 | 0.6519 | 8.9196 | 3.4403 | 1.93 |
| Spline | linear | partial | — | 0.6531 | 8.8849 | 3.4650 | 1.92 |
| | | full | 3.8461 | 0.6520 | 8.9240 | 3.5144 | 2.29 |
| | quadratic | partial | — | 0.6545 | 8.8973 | 3.3828 | 1.78 |
| | | full | **3.8453** | 0.6519 | **8.8776** | 3.4565 | 2.15 |
| Autoregressive | linear | partial | — | 0.6519 | 8.9177 | **3.3728** | 1.95 |
| | | full | 3.8481 | 0.6519 | 8.9161 | 3.4557 | 2.31 |
| | quadratic | partial | — | **0.6518** | 8.9095 | 3.4662 | 1.81 |
| | | full | 3.8454 | 0.6519 | 8.8820 | 3.4468 | 2.19 |

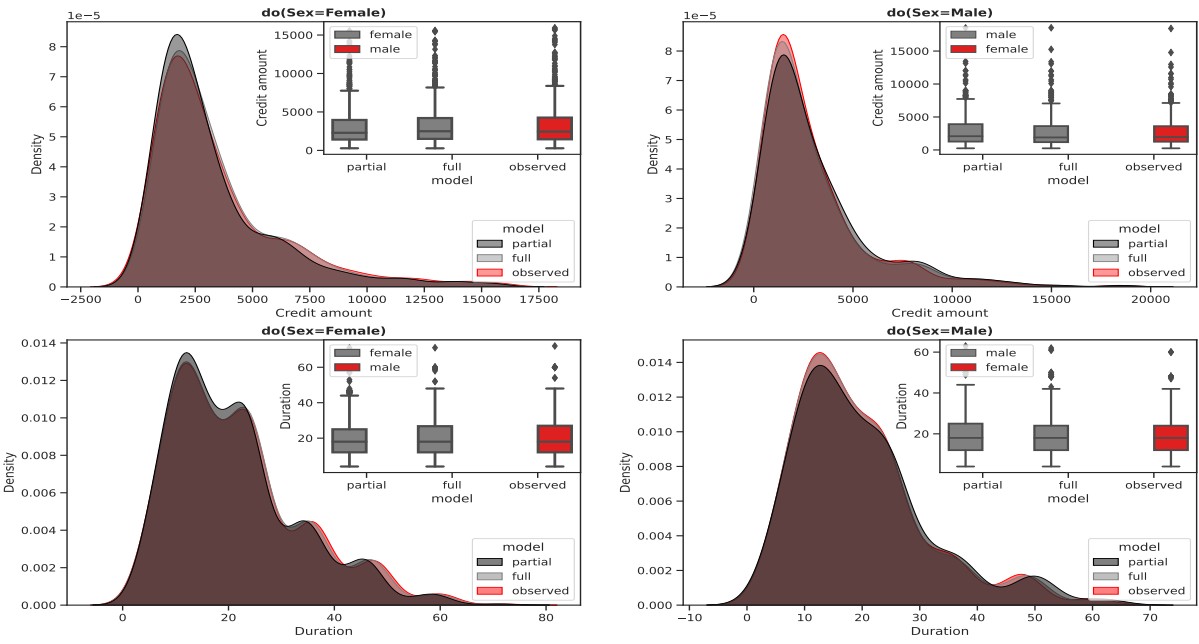

Figure 12: Flows' combination: spline, flow-order: linear

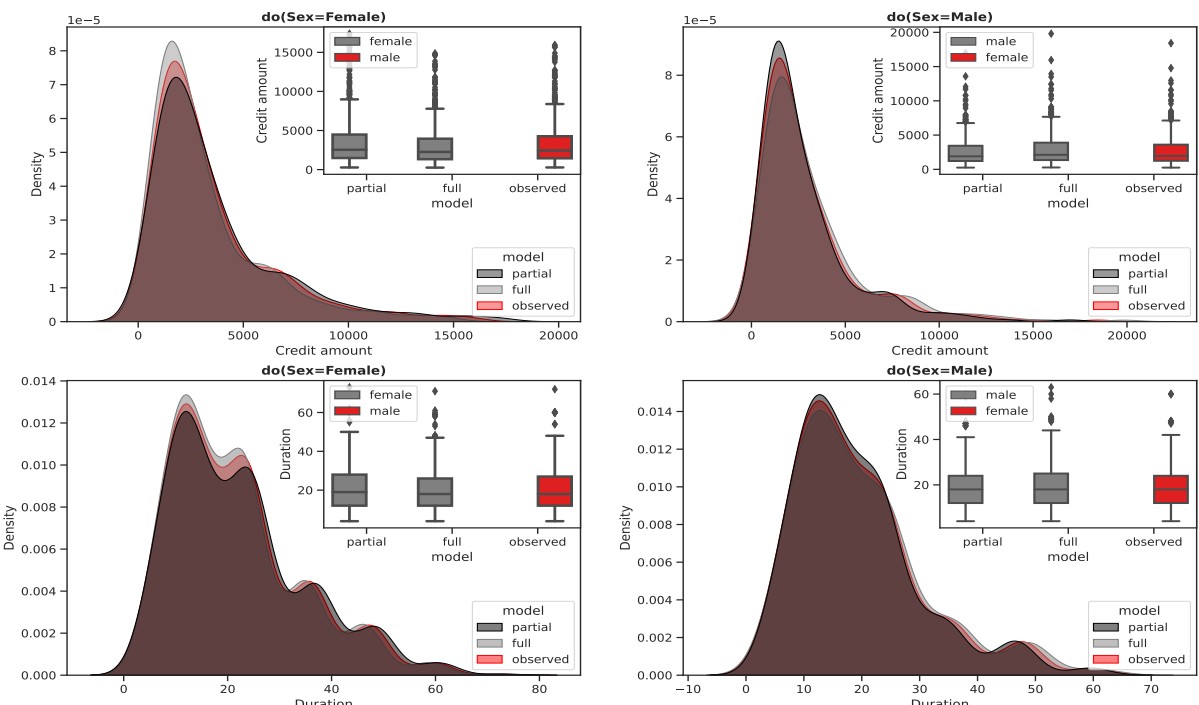

Figure 13: Flows' combination: affine, flow-order: quadratic

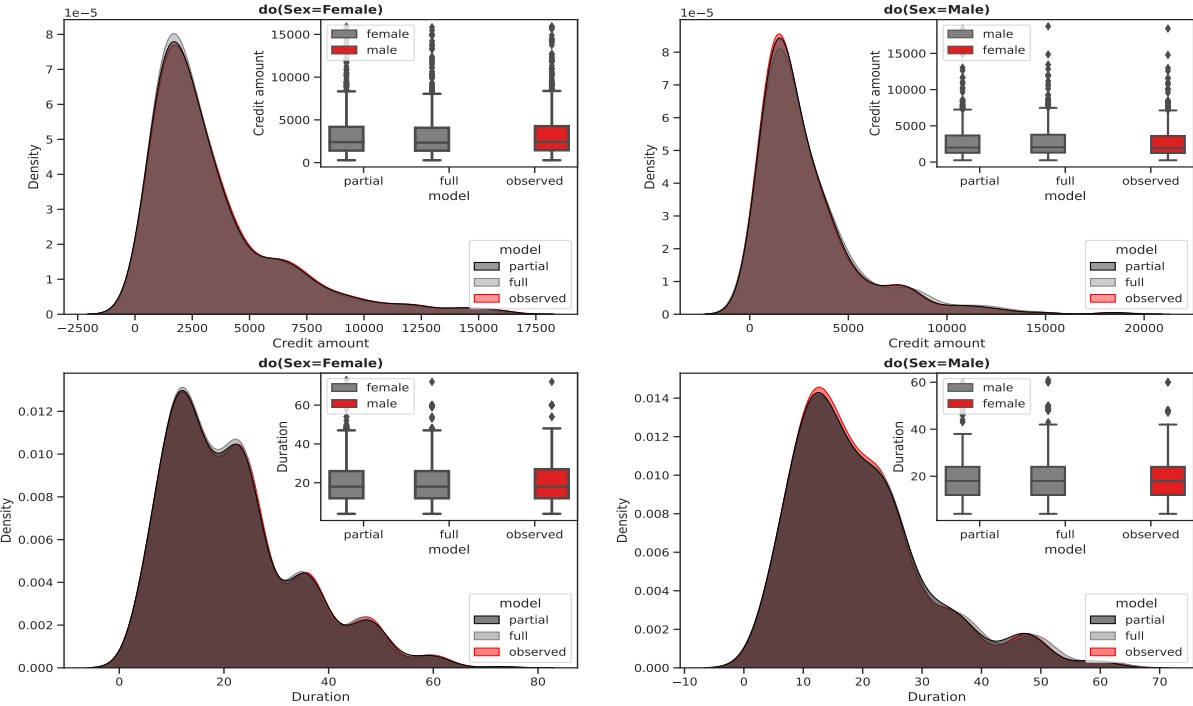

Figure 14: Flows' combination: autoregressive, flow-order: linear

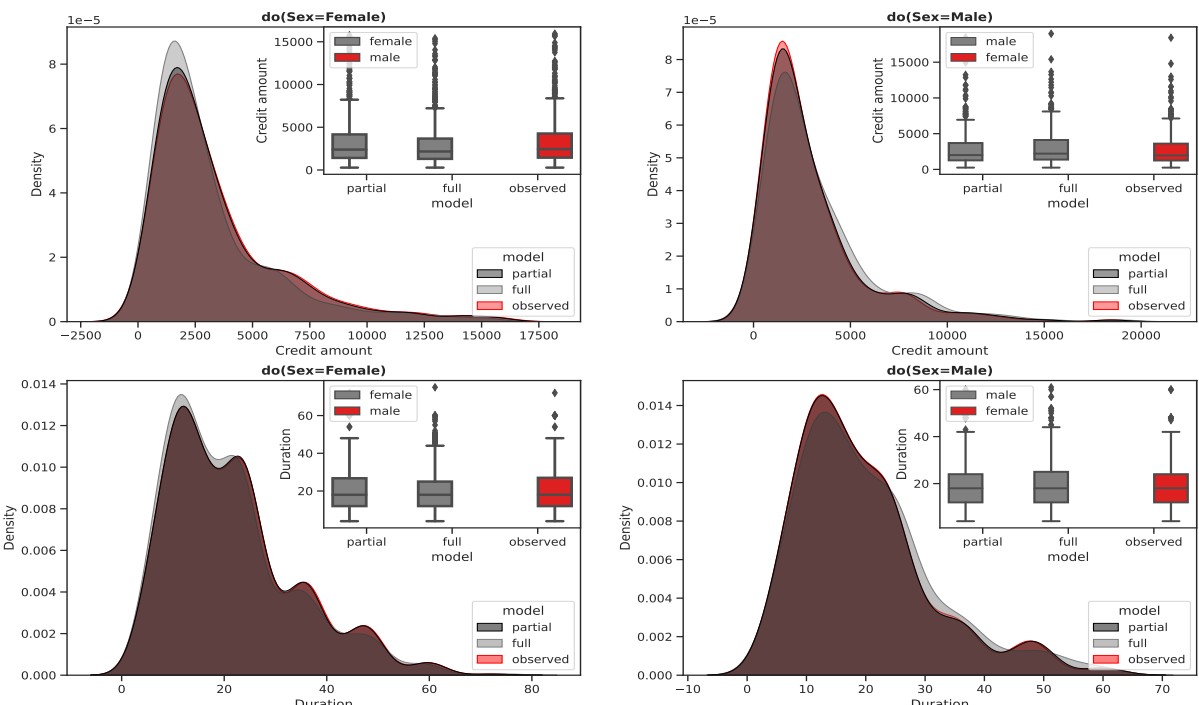

Figure 15: Flows' combination: autoregressive, flow-order: quadratic

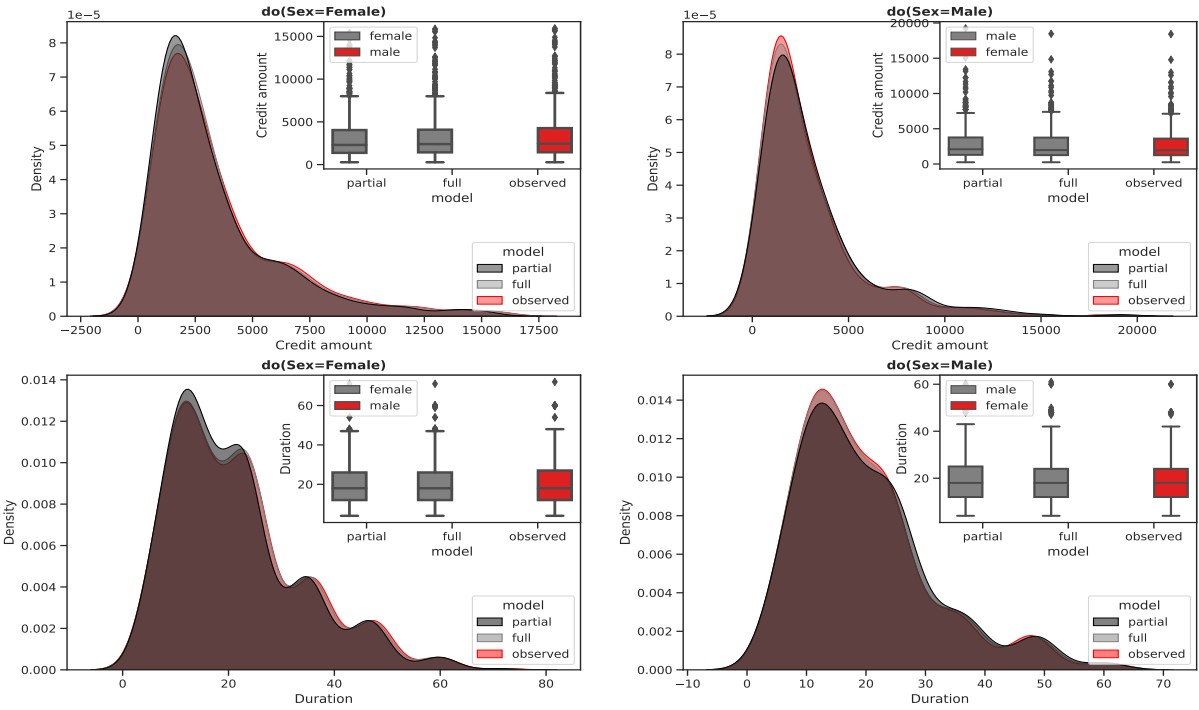

Figure 16: Flows' combination: spline, flow-order: quadratic

