# OpenReview forum: "On Noise Abduction for Answering Counterfactual Queries: A Practical Outlook "
_TMLR — Accepted by TMLR_

### Review · Reviewer_JWSz · 2022-08-10

**Summary Of Contributions:**

This paper studies the evaluation of counterfactual probabilities (i.e., counterfactual queries) in a fixed structural causal model (SCM), provided with its complete model parametrization. Specifically, the underlying SCM is a Markovian model that consists of observed variables (i.e., the query of interest) and unobserved variables (i.e., exogenous noises); no unobserved noise could affect multiple (more than one) observed variables simultaneously. That is, there is no unobserved confounder in the system. The authors further assume that the structural function determining values of every observed variable is invertible, parameterized as a normalizing flow. Existing counterfactual evaluation methods in Markovian model with invertible structural functions generally follow Pearl's three-step algorithm involving abduction, action, and prediction. All unobserved variables (exogenous) have to be abducted to update the posterior distribution given observed evidence. This step is typically computationally heavy. To address this issue, this paper shows that it may not be necessary to infer all unobserved variables in the SCM. They introduce a graphical criterion to determine a sufficient subset of unobserved variables for the abduction so that the counterfactual query could be properly evaluated in a fully specified Markovian model. In principle, this novel pre-abduction procedure could improve the computational efficiency of exiting counterfactual evaluation methods using normalizing flows.

**Broader Impact Concerns:**

The paper is mostly theoretical. Its longterm societal impact is not immediate to see.

**Requested Changes:**

1. Provide a precise definition of "essential exogenous noise" in the pre-abduction step.

2. Provide additional experiments comparing baseline algorithms that do not apply the proposed pre-abduction step. Report performance metrics for all algorithms, e.g., the training time.

3. (Optional) Reorganize the paper: start with the evaluation of atomic counterfactual queries; then introduce semi-soft\semi-hard interventions; and finally, the general counterfactual evaluation with soft and semi-soft interventions. However, I understand that the author might have challenges with the page limit. This request is optional.

**Strengths And Weaknesses:**

#### **Strength.** The proposed method is intuitive, and the proofs seem sound. Generally, one should abduct exogenous noise associated with all the ancestor nodes of the potential outcome, which is the target of the query. It is verifiable that such a collection of exogenous noises is sufficient for computing the target counterfactual probability. It is only a subset of all possible unobserved variables in the system.

#### **Weakness.** Overall, I think this paper proposes a practical technique to improve the computational efficiency of counterfactual evaluation methods based on normalizing flows. However, I do have some concerns, summarized below.

1. I agree with the proposed method's general idea and with most of the technical details. However, the proposed method's implementation is unclear and deserves further explanation. For instance, in the pseudo-code on Page 8, Step 1 asks one to identify all exogenous noises that are "essential" to answer a query $Q$. However, the concept of "essential exogenous noises" is not well defined. I can see that Theorems 1-4 are closely related to this definition. It would still be appreciated if the authors could provide a precise and formal definition.

2. In their initial work, Pawlowski et al. (2020) focused on Markovian models where UCs do not exist. It is somewhat surprising that the literature has not moved beyond the assumption of no UCs and considers semi-Markovian models where UCs generally exist. Note that the presence of UCs is arguably one of the most crucial challenges in modern causal inference. It could be great to see if one could apply the computational framework of normalizing flows to this more general class of causal models. Could the authors comment on this? What are the main challenges in applying counterfactual evaluation with invertible functions in semi-Markovian models?

3. The primary motivation of this paper is to reduce the computational cost when evaluating counterfactual queries from a fully specified SCM. However, in the experiments, there is no baseline algorithm for comparing with the proposed method. It would be interesting to know whether the proposed method reduces the computational cost of counterfactual evaluation methods like (Pawlowski et al. 2020); if so, by how much?

4. This might be a minor one. The organization of the paper could be improved. The idea of semi-soft\semi-hard intervention is interesting. It is similar to the counterfactual randomization introduced in (Forney, Bareinboim & Pearl, 2015). However, it seems orthogonal to the central challenge studied in this paper: the selection of exogenous variables for the abduction. For clarity, I would recommend first considering the counterfactual evaluation for a standard atomic query (e.g., $P(Y_{x=0}, Y_{x=1})$), and then introducing semi-soft\semi-hard interventions as a generalization.

---

> ### Author Response · Authors · 2022-09-01
> **Reply to Reviewer JWSz**
>
> Thank you for putting the summary of contributions in so nicely. We are glad that you find the pre-abduction novel. We are encouraged that you find our proposed method to be intuitive. We would like to address your concerns one by one as follows:
>
> ---
>
> (W1). **I agree with the proposed method’s general idea and with most of the technical details. However, the proposed method’s implementation is unclear and deserves further explanation. For instance, in the pseudo-code on Page 8, Step 1 asks one to identify all exogenous noises that are “essential” to answer a query . However, the concept of ”essential exogenous noises” is not well defined. I can see that Theorems 1-4 are closely related to this definition. It would still be appreciated if the authors could provide a precise and formal definition.**
>
> (RC1). **Provide a precise definition of “essential exogenous noise” in the preabduction step.**
>
>
> >  We have added the definition of ‘sufficient’ and ‘essential’ sets of exogenous noises (Section 5, Definition 2). Any superset of the essential set of noises is sufficient. We have shown that set of noises attached to variables that lie on paths from intervened set to the target set in the modified graph is essential (Section 5, Theorem 5).
>
> ---
> (W2). **In their initial work, Pawlowski et al. (2020) focused on Markovian models where UCs do not exist. It is somewhat surprising that the literature has not moved beyond the assumption of no UCs and considers semi-Markovian models where UCs generally exist. Note that the presence of UCs is arguably one of the most crucial challenges in modern causal inference. It could be great to see if one could apply the computational framework of normalizing flows to this more general class of causal models. Could the authors comment on this? What are the main challenges in applying counterfactual evaluation with invertible functions in semi-Markovian models?**
>
> >  We agree that UC is one of the most crucial challenges in modern causal inference. Identifiability of counterfactuals is a big concern in such settings. Another challenge is immediate: If $U$ is unobserved, how do you estimate $f_{U}$ ? Normalizing flows transform a simple probability distribution into the target (observed data) distribution. This is an exciting problem and not within the scope of the current paper. We plan to explore this direction for future work.
>
> ---
> (W3). **The primary motivation of this paper is to reduce the computational cost when evaluating counterfactual queries from a fully specified SCM. However, in the experiments, there is no baseline algorithm for comparing with the proposed method. It would be interesting to know whether the proposed method reduces the computational cost of counterfactual evaluation methods like (Pawlowski et al. 2020); if so, by how much?**
>
> (RC2). **Provide additional experiments comparing baseline algorithms that do not apply the proposed pre-abduction step. Report performance metrics for all algorithms, e.g., the training time.**
>
>
> >  It is incorrect that “in the experiments, there is no baseline algorithm for comparing with the proposed method.” Please note that - in both synthetic data (Section 6.1) and real data experiment (Section 6.2), the partial model applies pre-abduction, whereas the full model does not apply pre-abduction. In both synthetic and real data experiments, we have implemented Pawlowski et al. 2020’s framework of counterfactual estimation using normalizing flows.
>
>
> >  Thank you for the suggestion of reporting training time. We have experimented and reported the training times. (Table 1 and Figure 4 in the main paper and Table 2 in the appendix D)
>
> ---
> (W4). **This might be a minor one. The organization of the paper could be improved. The idea of semi-soft\ semi-hard intervention is interesting. It is similar to the counterfactual randomization introduced in (Forney, Bareinboim & Pearl, 2015). However, it seems orthogonal to the central challenge studied in this paper: the selection of exogenous variables for the abduction. For clarity, I would recommend first considering the counterfactual evaluation for a standard atomic query (e.g., $P(Y_{x}=0,Y_{x}=1)$ ), and then introducing semi-soft\ semi-hard interventions as a generalization.**
>
> (RC3)**(Optional) Reorganize the paper: start with the evaluation of atomic counterfactual queries; then introduce semi-soft\semi-hard interventions; and finally, the general counterfactual evaluation with soft and semi-soft interventions. However, I understand that the author might have challenges with the page limit. This request is optional.**
>
> > Semi-soft \semi-hard intervention is motivated by Example 1 (Section 3, Page 5). Therefore, the flow may be hampered if semi-soft\semi-hard is introduced after discussing the atomic query. The page constraint is another issue with adding more text.
>
> `'W' stands for weakness and 'RC' stands for requested changes `

---

### Review · Reviewer_WGHU · 2022-08-18

**Summary Of Contributions:**

Counterfactual inference is a fundamental problem in causality. Recently, some related studies are proposed towards it following the three-step framework (abduction, action, prediction) with some novel deep learning techniques. In the abduction step, the previous methods infer all noises, which are with high computational costs. In this paper, the authors propose that it is not necessary to infer all noises. According to the causal graph which is pre-known, they present the method to infer part of noises which are sufficient for achieving the counterfactual inference.

**Broader Impact Concerns:**

No.

**Requested Changes:**

I cannot give some detailed technical suggestions right now, because it is hard for me to evaluate the method due to the unsatisfactory writing. For example, the main proposed process including the four steps is shown on Page 8, but I am not sure how the authors define **essential exogenous noises** and how to find the essential exogenous noises. And the **projection operator depends on** $do(A\leftarrow a)$ is also ambiguous. I can guess their implications, but I cannot be very confident. I suggest the authors revise the writing carefully at first. I first present the questions which I think the authors should address in the paper. Then I give some suggestions.


Questions:

1. What are the strict definitions of essential exogenous noises, projection operator depends on $do(A\leftarrow a)$?

2. "Theorem 4 allows us to intervene on the variables outside..." What is the benefit of intervening on the variables outside...? I guess that the authors want to say when we intervene on some variables, the noise of these variables can be ignored. Thus we hope to have an expression with more variables intervened, in which way we do not need to infer many noises. Is it right?

3. Could the authors explain the difference between Thm.1 and Thm.2? I think they can be merged.


4. Section 2.2.1: Consider the generating assignment $x=f(\epsilon)$, if $f$ is a invertible function, does it necessarily mean that the distribution $P(\epsilon|X=x^{obs})$ is identifiable?

5. What does it mean by $2$ in "In a causal model with joint distribution having joint density satisfying 2"?

6. What does it mean by "Standard tools of the SCM framework don’t inherently restrict intervention. one could at least in theory intervene unconditionally on any subset of variables to perform counterfactual analysis."? Even if we have the interventional distribution, it is not necessarily that the counterfactual is identifiable, as shown by Peters et. al 2017.

7. What is the connection between Section 3 and Section 5? Or what role does semi-hard intervention play in this paper?

Suggestions:
1. It is not proper to say "This paper tackles problem of identifying exogenous noises that must be abducted for counterfactual inference.". Are there any proofs to show that these exogenous noises **must be abducted** for counterfactual inference? As far as I see, if we infer these noises, we can achieve the counterfactual inference. But I am not sure whether the counterfactual inference is impossible if we do not infer these noises.

2. It is not quite proper to have no introduction to normalizing flows but just say a sentence "see Papamakarios et al. (2019)."

3. The expression of Thm.2 is not proper in an academic paper. The expression such as "$X_j$ is affected in 'counterfactual'" is too sloppy.

4. In Page 3, it seems that there is a typo. ' Read: “The value of $X_i$ in situation $\epsilon$, had $X_j$ been $x_j$” ' or ' Read: “The value of $X_i$ in situation $\epsilon$, had $X_j$ been $x_j'$” '.

5. The sentence "In general, it is not immediately clear how to design effective experimental procedures for evaluating counterfactuals, or how to compute them from observational data" is not quite exact. It should not be "not immediately clear". The counterfactual is not necessarily identifiable given both observational distribution and interventional distribution in general, thus there should not exist a method to design effective experimental procedures for evaluating counterfactuals or compute them from observational data for general cases. Some examples to indicate the unidentifiability are given in Peters et. al 2017.

6. The definitions for unit-level and population-level counterfactual seem to be missing.

7. Caption of Fig.1 1: causal graph -> causal graphs, and a missing full stop.

8. I suggest the authors change the title of Section 3. In this part, the authors only introduce the definition of semi-soft intervention. The title could be more clear.

9. The definition of **path** in Section 4 seems wrong. It seems that any of $X_{i_1},\cdots,X_{i_{m-1}}$ is adjacent to $X_{i_m}$.

10. "Every SCM entails a unique joint distribution over the variables $X = (X1, ...,Xp)$ such that relationships in (1) hold true.: "such that relationships in (1) hold true" seems to be redundant and not exact here. There is an SCM at first, then there is observational data based on the distribution of noise.

11. In the last paragraph of the introduction:"our work shows that it mayn’t be necessary to infer all the noise variables in the SCM and identifies exogenous noise variables that we must infer...": what do the authors want to express by "that we must infer". It seems to has the same implication as "it mayn’t be necessary to infer all...".

**Strengths And Weaknesses:**

Advantage:

The deep learning techniques introduced to counterfactual inference are favored.

Disadvantage:
1. The contribution is limited.
2. The writing can be improved largely. Although the idea is simple and there are not many new technical contributions, the paper is very hard to read because there are so many unclear and distracting expressions. I show them in detail in the next part. Due to the unsatisfactory writing, it is hard for me to evaluate the proposed method with high confidence or present some technical suggestions.

---

> ### Author Response · Authors · 2022-09-01
> **Reply to Reviewer WGHU (1/2)**
>
> We thank you for the detailed review, feedback, and encouragement. We would like to address your concerns one by one as follows:
>
> ---
>
> (Q1). **What are the strict definitions of essential exogenous noises, projection operator depends on $do(\mathcal{A}\leftarrow a)$**
>
> (S1). **It is not proper to say “This paper tackles problem of identifying exogenous noises that must be abducted for counterfactual inference.”. Are there any proofs to show that these exogenous noises must be abducted for counterfactual inference? As far as I see, if we infer these noises, we can achieve the counterfactual inference. But I am not sure whether the counterfactual inference is impossible if we do not infer these noises.**
>
> (S11). **In the last paragraph of the introduction“our work shows that it mayn’t be necessary to infer all the noise variables in the SCM and identifies exogenous noise variables that we must infer...”: what do the authors want to express by ”that we must infer”. It seems to has the same implication as ”it mayn’t be necessary to infer all...”.**
>
> > We have added the definition of ‘sufficient’ and ‘essential’ sets of exogenous noises (Section 5, Definition 2). Any superset of the essential set of noises is sufficient. We have shown that set of noises attached to variables lie on paths from intervened set to the target set in the modified graph is essential (Section 5, Theorem 5).
>
> > Projection operator $\pi_{\mathcal{A}}$ maps $\epsilon$ to $\epsilon_{\mathfrak{p}}$ where  {$ \epsilon_{i}$}$_{i\in \mathfrak{p}}$ is essential set of exogenous noises to $\mathcal{Q}$ (Section 5, page 8).
>
> ---
>
> (Q2). **“Theorem 4 allows us to intervene on the variables outside...” What is the benefit of intervening on the variables outside...? I guess that the authors want to say when we intervene on some variables, the noise of these variables can be ignored. Thus we hope to have an expression with more variables intervened,
> in which way we do not need to infer many noises. Is it right?**
>
> > Yes.
>
> ---
>
> (Q3). **Could the authors explain the difference between Thm.1 and Thm.2? I think they can be merged.**
>
> > They could have been merged since both originate from observation 1, or Theorem 2 could have been interpreted as Lemma. But we prefer to keep them separate because - theorem 1 guarantees that  variables except the intervention set and their descendants will not change from observed value in counterfactuals. Moving ahead, Theorem 2 ensures we may see a change in a variable $X_{i}$  from observed value in counterfactuals iff  at least one of $X_{i}$'s ancestors has been intervened.  Yes, they are telling somewhat similar things but from two different perspectives. Theorem 1 suggests focusing on the intervention set and its descendants, but it was not immediate to us that we could restrict the search on  paths from the intervention set to the target set of the query.
> Theorem 2 suggests looking at only the query's target set and its ancestors.  Combining them, we can constrain the search space for noise variables.
>
> ---
>
> (Q4). **Section 2.2.1: Consider the generating assignment $x=f(\epsilon)$, if  is a invertible function, does it necessarily mean that the distribution $P(\epsilon|X=x^{obs})$ is identifiable?**
>
> > If $f$ is known, $\mathbb{P}(\epsilon|X=x^{obs})$ is identifiable. In reality, $f$ is unknown, so we need to estimate $f$. Thus, the assumption on $\mathbb{P}(\epsilon)$ is necessary for the estimation in the very first place.
>
> ---
>
> (Q5). **What does it mean by  in "In a causal model with joint distribution having joint density satisfying 2"?**
>
> > We tried to mean a causal markov model, but since it was confusing, we have rewritten the identifiability subsection (2.2.1).
>
> ---
>
>
> (Q6). **What does it mean by ”Standard tools of the SCM framework don’t inherently restrict intervention. one could at least in theory intervene unconditionally on any subset of variables to perform counterfactual analysis.”? Even if we have the interventional distribution, it is not necessarily that the counterfactual is identifiable, as shown by Peters et. al 2017.**
>
> > In this paragraph, we tried to highlight the scope of interventions in SCM rather than the identifiability issue. Once we assume structural assignments are parameterized by normalizing flow, counterfactuals are identifiable. Next, we wanted to discuss the scope of interventions for performing counterfactual analysis. Since it was confusing, we have introduced it under a new subsection (2.2.2).
>
>
>
> `'Q' stands for question and 'S' stands for suggestion. `

---

> ### Author Response · Authors · 2022-09-01
> **Reply to Reviewer WGHU (2/2)**
>
> (Q7). **What is the connection between Section 3 and Section 5? Or what role does semi-hard intervention play in this paper?**
>
> >Semi-soft\semi-hard intervention is a specific type of soft intervention (or mechanism change)  for which we do not need to infer the exogenous parent of the intervened variable (much like hard intervention).  In example 1, case (b) $do(Y=X+\epsilon_{Y})$ is an example of soft-intervention  and  case (c) $do(Y=Y+2)$ is an example of semi-soft intervention. In example 1, in  case (b), inferring $\epsilon_{Y}$ is a must. But in  case (c),  It is not required to infer $\epsilon_{Y}$  like the hard intervention case in  case (a).
>
> ---
>
> (S2). **It is not quite proper to have no introduction to normalizing flows but
> just say a sentence ”see Papamakarios et al. (2019).”**
>
> > Due to the page constraint, we have added a section introducing normalizing
> flows in  appendix A.
>
> ---
>
> (S3). **The expression of Thm.2 is not proper in an academic paper. The expression such as " $X_{j}$ is affected in 'counterfactual'" is too sloppy.**
>
> > We have restated Thm.2 to address this concern.
>
>
>
> ---
>
> (S4). **In Page 3, it seems that there is a typo. ' Read: “The value of  $X_{i}$ in situation $\epsilon$, had $X_{j}$ been $x_{j}$ ” ' or ' Read: “The value of $X_{i}$ in situation $\epsilon$ , had $X_{j}$ been
> $x_{j}'$” '**
>
> (S7). **Caption of Fig.1 1: causal graph $\rightarrow$ causal graphs, and a missing full stop.**
>
> (S9). **The definition of path in Section 4 seems wrong. It seems that any of $X_{i_{1}},X_{i_{2}},...,X_{i_{m}}$  is adjacent to $X_{i_{m}}$**
>
> > Thanks for pointing this out. We have fixed these errors. We have proofread the complete paper carefully and tried to correct all the typos and mistakes.
>
> ---
>
> (S5). **The sentence “In general, it is not immediately clear how to design effective experimental procedures for evaluating counterfactuals, or how to compute them from observational data” is not quite exact. It should not be ”not immediately clear”. The counterfactual is not necessarily identifiable given both observational distribution and interventional distribution in general, thus there should not exist a method to design effective experimental procedures for evaluating counterfactuals or compute them from observational data for general cases. Some examples to indicate the unidentifiability are given in Peters et. al 2017.**
>
> > We agree. We have rewritten the identifiability subsection (2.2.1).
>
> ---
>
> (S6). **The definitions for unit-level and population-level counterfactual seem to be missing.**
>
> > The definition of unit-level counterfactual was already given in section 2.2. However, we have extended the idea of population-level counterfactuals. The distinction between unit-level and population-level counterfactual is whether we are talking about a  particular situation (or individual) $\epsilon$ or all the situations (or population) (sometimes on average). In literature, we find definitions of Natural direct effect, Natural indirect effect, etc... These are examples of population-level counterfactuals. We are unaware of formal definition of population-level counterfactuals. It should be clear from semantic understanding and from the context.
>
> ---
>
> (S8). **I suggest the authors change the title of Section 3. In this part, the authors only introduce the definition of semi-soft intervention. The title could be more clear.**
>
> > In section 3, in addition to the definition of semi-soft intervention, we also have provided examples of counterfactuals with different types of intervention (Example 1), which is the base of our paper.
>
>
> ---
>
> (S10). **``Every SCM entails a unique joint distribution over the variables $X=(X_{1},X_{2},...,X_{p})$ such that relationships in (1) hold true.: "such that relationships in (1) hold true" seems to be redundant and not exact here. There is an SCM at first, then there is observational data based on the distribution of noise.**
>
> >Thanks for the suggestion. We have omitted the  "such that relationships in (1) hold true" part.
>
>
> `'Q' stands for question and 'S' stands for suggestion. `

---

### Review · Reviewer_uvek · 2022-08-19

**Summary Of Contributions:**

The main observation of this paper is that for many counterfactual queries, we don't need to perform the abduction step (inference over the exogenous noise variables), so given a query, we can just do inference over the necessary variables rather than the full set of noise variables.

**Broader Impact Concerns:**

No concerns.

**Requested Changes:**

* Can you give an example of a task that is not feasible unless you restrict the set of variables for the abduction step? Or at least show a significant computational saving empirically?

* Better experiments mentioned above - error bars, report computational costs, etc.

* This paper needs a serious proof read for grammatical errors and typos, etc.. Normally I don't make a big deal out typos, grammar, etc., but in this paper some of the sections are relatively well written, while others are full of typos and grammar errors. This suggests that the authors just didn't bother to do a proper proof read before submitting - which is very sloppy: the reviewing process is not meant as a substitute for proof reading. Here's the errors that I wrote down (a subsets of all errors):
    - Page 2 - "mayn’t" -> "may not"
    - Page 4 - "one could at least" -> "One could at least"
    - Page 2 - "and semi-soft(semi-hard)" -> "and semi-soft (semi-hard)" [missing space]
    - Page 6 - "In-fact, for the sake of keeping things simple, we take resort of this for rest" -> "In fact" and reword "we take resort of this for rest..."
    - Page 7 - reword Theorem 2


**Strengths And Weaknesses:**

Strengths:
* The observation the not all noise variables are relevant to the counterfactual query makes sense.
* I liked example 1 which is used to support this observation. Counterfactuals are not well-understood by a lot of the machine learning community, so a series of examples of queries and the corresponding inference steps serves a useful pedagogical role.

Weaknesses:
* While I know I'm not meant to be reviewing for impact - I had a really hard time thinking of settings where this idea would be used. If you're using a flow based model to estimate your SCM, the abduction step is just a forward pass through the inverse model so this save you having to query a couple of the outputs, but surely this is a negligible computational saving?
* It's not clear what semi-soft / semi-hard interventions add - they don't seem to be treated differently in the theory?
* Given that the main claim is computational efficiency, you should be reporting the computational saving - e.g. what is the difference in wall-clock time for inference for the two approaches across a large number of seeds?
* The differences between the full model and the partial model in the experiments were mostly as expected (they're mathematically equivalent, so we should expect big differences beyond finite sample error), with the exception of figure 3 b. I didn't understand why the full model had much larger errors. Is this just an unlucky seed?
* The synthetic experiments should have been run multiple times with different seeds to report standard errors.

---

> ### Author Response · Authors · 2022-09-01
> **Reply to Reviewer uvek (1/2)**
>
> Thank you for the positive and insightful feedback! We are encouraged that you find our observation makes sense, like our example 1. We would like to address your concerns one by one as follows:
>
> ---
>
> (W1). **While I know I’m not meant to be reviewing for impact - I had a really hard time thinking of settings where this idea would be used. If you’re using a flow based model to estimate your SCM, the abduction step is just a forward pass through the inverse model so this save you having to query a couple of the outputs, but surely this is a negligible computational saving?**
>
> > Counterfactual techniques are being integrated with deep learning very recently. A lot of work is yet to be done. So we might give you some loose ideas but not the very concrete ones - If nodes of the causal graph are high dimensional, for example, images (see causal structure in Geiger and Sauer (2021)), pre-abduction will turn out to be really useful as at the very core, it reduces neural networks from a model. If these neural network architectures are heavy, it will save considerable time. Even recently, neural networks have been treated as SCM (see Chattopadhyay et al. (2019)). It would be interesting to see the role of pre-abduction in answering counterfactual queries there.
> Computational time saving is not the only takeaway from pre-abduction.
> We would like to point out the other aspects.
> > - Loosely speaking, if you have 10-20 nodes in a causal graph (which is common, e.g., a causal graph for the UCI adult dataset), to answer a counterfactual query without pre-abduction, you need to model normalizing flows for each variable. Designing 10-20 flows (one for each of the variables in the causal graph) and jointly optimizing them is difficult and may not be required.  If one of the flows doesn’t approximate the target density well enough, then the error will flow through the descendants. In this regard, pre-abduction helps to prioritize and reduce the number of flows we need to model and estimate.
> > - We can relax assumptions on noise that are not essential for the targeted counterfactual query.
> > - Since we only use normalizing flows to parameterize the structural assignments for identifiability (and not for identifying essential exogenous noises), Pre-abduction can be applied to other counterfactual estimation frameworks as long as counterfactuals are identifiable.
>
> > **References:**
>
> > A. Geiger and A. Sauer. Counterfactual generative networks. In Interna-
> tional Conference on Learning Representations (ICLR), 2021
>
> >A. Chattopadhyay, P. Manupriya, A. Sarkar, and V. N. Balasubramanian.
> Neural network attributions: A causal perspective. In Proceedings of the 36th
> International Conference on Machine Learning, volume 97 of Proceedings of
> Machine Learning Research, pages 981–990. PMLR, 09–15 Jun 2019
>
> ---
>
> (W2). **It’s not clear what semi-soft / semi-hard interventions add - they don’t
> seem to be treated differently in the theory?**
>
> >Semi-soft\semi-hard intervention is a specific type of soft intervention (or mechanism change)  for which we do not need to infer the exogenous parent of the intervened variable (much like hard intervention).  In example 1, case (b) $do(Y=X+\epsilon_{Y})$ is an example of soft-intervention  and  case (c) $do(Y=Y+2)$ is an example of semi-soft intervention. In example 1, in  case (b), inferring $\epsilon_{Y}$ is a must. But in  case (c),  It is not required to infer $\epsilon_{Y}$  like the hard intervention case in  case (a).
>
> ---
>
> (w3). **Given that the main claim is computational efficiency, you should be reporting the computational saving - e.g. what is the difference in wall-clock time for inference for the two approaches across a large number of seeds?**
>
>
> > Inference time isn't significant in our case since it's just a single pass through the flow-SCM. However, we have experimented and reported the training times and ratio of training times. (Table 1 and Figure 4 in the main paper and Table 2 in the appendix D)
>
> ---
>
> (W4). **The differences between the full model and the partial model in the experiments were mostly as expected (they’re mathematically equivalent, so we should expect big differences beyond finite sample error), with the exception of figure 3b. I didn’t understand why the full model had much larger errors. Is
> this just an unlucky seed?**
>
> (W5). **The synthetic experiments should have been run multiple times with different seeds to report standard errors.**
>
> (RC2). **Better experiments mentioned above - error bars, report computational costs, etc**
>
>
> > Thank you for the suggestions. We have incorporated these suggestions into our experiments.
> ---
>
> `'W' stands for weakness and 'RC' stands for requested changes `

---

> ### Author Response · Authors · 2022-09-01
> **Reply to Reviewer uvek (2/2)**
>
> (RC1). **Can you give an example of a task that is not feasible unless you restrict
> the set of variables for the abduction step? Or at least show a significant computational saving empirically?**
>
> > In principle, any sufficient set of exogenous noise will make the task of counterfactual prediction feasible. Please see figure 4 and table 1 (in the revised submission) under the synthetic data experiment for significant computational time savings.
>
> ---
>
> (RC3). **This paper needs a serious proof read for grammatical errors and typos, etc.. Normally I don't make a big deal out typos, grammar, etc., but in this paper some of the sections are relatively well written, while others are full of typos and grammar errors. This suggests that the authors just didn't bother to do a proper proof read before submitting - which is very sloppy: the reviewing process is not meant as a substitute for proof reading. Here's the errors that I wrote down (a subsets of all errors):**
>
> * **Page 2 - "mayn’t" $\rightarrow$ "may not"**
>
> * **Page 4 - "one could at least" $\rightarrow$ "One could at least"**
>
> * **Page 2 - "and semi-soft(semi-hard)" $\rightarrow$ "and semi-soft (semi-hard)" [missing space]**
>
> * **Page 6 - "In-fact, for the sake of keeping things simple, we take resort of this for rest" $\rightarrow$ "In fact" and reword "we take resort of this for rest..."**
>
> * **Page 7 - reword Theorem 2****
>
>
> > Thank you for giving a thorough review. We have incorporated all the requested changes. . We have fixed these errors. We have proofread the complete paper carefully and tried to correct all the typos and mistakes.
>
>
> `'W' stands for weakness and 'RC' stands for requested changes `

---

### Decision · Action_Editors · 2022-09-26

**Recommendation:** Accept with minor revision

**Comment:**

This paper studies a nice improvement on Pearl's three-step algorithm for evaluating counterfactual queries in a fully specified causal model. This improvement reduces the total number of exogenous variables to be updated during the abduction phase, thus reducing the total training time. Simulation results show that the proposed method is able to reduce the training time by half compared to the standard approach which updates the full model. Therefore, this paper provides a compact solution that improves the computational efficiency of a popular algorithm in causal inference. In general, the proposed solution is theoretically sound and demonstrates good properties. However, there are some important aspects that can be improved. In definition 2, there is not a strict definition for "Q can be answered by inferring \bar{\epsilon} only". The authors should emphasize in Def. 2 that "Q can be answered by inferring \bar{\epsilon} only by the three-step algorithm". Without the limitation "by the three-step algorithm", the proof of Theorem 5 seems incorrect. One important question here is that is it possible to identify the counterfactual by other methods instead of the three-step algorithm? I cannot see the impossibility by the current proofs. It seems it is possible that although we cannot infer, we can identify the counterfactual queries by other methods, in which case the set can also be essential to Q. Hence, I suggest the authors write the definition 2 more carefully, which can influence the correctness of their result. Thus, I would like to recommend accepting the submission with minor revision.

---

> ### Author Response · Authors · 2022-10-23
> **Camera-ready version**
>
> Dear reviewers and AE,
>
> Thank you for the time and effort invested in the review process. It helped us to improve the quality of the paper. We have just uploaded the de-anonymized camera-ready version. The uploaded camera-ready version now reflects the requested changes.
>
> Best Regards,
> The Authors